# Mind the Gap Between Prototypes and Images in Cross-domain Finetuning

**Hongduan Tian[1], Feng Liu[2], Zhanke Zhou[1], Tongliang Liu[3], Chengqi Zhang[4], Bo Han[1]†**

[1]TMLR Group, Department of Computer Science, Hong Kong Baptist University
[2]TMLR Group, University of Melbourne    [3]Sydney AI Center, The University of Sydney
[4]Department of Data Science and Artificial Intelligence, The Hong Kong Polytechnic University
`{cshdtian, cszkzhou, bhanml}@comp.hkbu.edu.hk`,
`fengliu.ml@gmail.com, tongliang.liu@sydney.edu.au, chengqi.zhang@polyu.edu.hk`

## Abstract

In *cross-domain few-shot classification* (CFC), recent works mainly focus on adapting a simple transformation head on top of a frozen pre-trained backbone with few labeled data to project embeddings into a task-specific metric space where classification can be performed by measuring similarities between image instance and prototype representations. Technically, an *assumption* implicitly adopted in such a framework is that the prototype and image instance embeddings share the same representation transformation. However, in this paper, we find that there naturally exists a gap, which resembles the modality gap, between the prototype and image instance embeddings extracted from the frozen pre-trained backbone, and simply applying the same transformation during the adaptation phase constrains exploring the optimal representations and shrinks the gap between prototype and image representations. To solve this problem, we propose a simple yet effective method, *contrastive prototype-image adaptation* (CoPA), to adapt different transformations respectively for prototypes and images similarly to CLIP by treating prototypes as text prompts. Extensive experiments on Meta-Dataset demonstrate that CoPA achieves the *state-of-the-art* performance more efficiently. Meanwhile, further analyses also indicate that CoPA can learn better representation clusters, enlarge the gap, and achieve minimal validation loss at the enlarged gap. The project repository of CoPA is available at: https://github.com/tmlr-group/CoPA.

## 1 Introduction

Cross-domain few-shot classification [49] (a.k.a. CFC) aims at learning to perform classification on tasks sampled from previously unseen domains with only a few labeled data. Recent works [12, 34, 29, 30], which aims at fast adapting a light model on top of the frozen pre-trained backbones for feature selection or transformation, has shown impressive generalization capability on CFC tasks.

Typically, URL [29], a representative work of this genre, proposes to fast fine-tune a linear transformation head on top of a frozen pre-trained backbone with the nearest centroid classifier loss [47]. The ultimate goal of URL is to project embeddings into a space where classification can be performed via measuring the similarity between representations of prototypes and image instances. Technically, an *assumption* implicitly adopted in URL is that the prototype and image instance embeddings share the same representation transformation (see the upper subfigure of Fig. 2) during the adaptation phase. Intuitively, however, prototype and image instance embeddings depict different levels of information. Specifically, the embeddings of an image instance encode the *instance-level* information from the given image, while the embeddings of a prototype contain some abstract and *higher-level* information that is commonly shared among image instances within the corresponding class yet discriminative from other classes. Thus, applying the same representation transformation as URL might constrain models from exploring the optimal representations respectively for prototypes and image instances.

---

†Correspondence to Bo Han (bhanml@comp.hkbu.edu.hk).

38th Conference on Neural Information Processing Systems (NeurIPS 2024).

In this paper, we first conduct an analysis to validate the intuition that prototypes and image instances describe different levels of information. Specifically, following Liang et al. [32], we visualize the distributions of prototype and instance embeddings extracted from a frozen pre-trained backbone and measure the Euclidean distance between the centroids of the normalized prototype and image instance embeddings. According to Fig. 1(a), the UMAP [37] visualization indicates that there *naturally* exists a gap, which resembles the modality gap demonstrated by Liang et al. [32] in visual language models (VLMs, e.g. CLIP [42]), between prototype and image instance embeddings. However, after applying the same representation transformation to both prototype and image instance embeddings as done in URL [29], we observe that the gap is numerically shrunken (see Fig. 1(b)). Moreover, by visualizing the

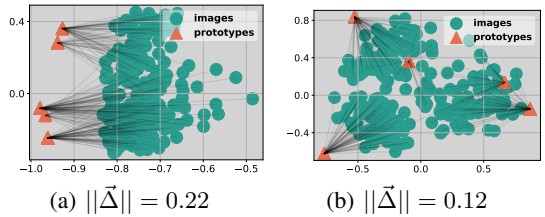

(a) $||\vec{\Delta}|| = 0.22$      (b) $||\vec{\Delta}|| = 0.12$

Figure 1: **There *naturally* exists a gap between prototype and image instance embeddings, but applying the same transformation shrinks such a gap.** Fig.(a) shows that there naturally exists a gap, which resembles the "modality" gap in visual language models, between prototype and image instance embeddings extracted from a frozen pre-trained backbone. However, Fig.(b) shows that the gap between the representations of prototypes and image instances is shrunk after applying the same representation transformation to both the image instance and prototype embeddings.

cluster distribution of data samples (see Figs. 3(b) and 3(c)), we further find that applying the same representation transformation to both prototype and image instance embeddings may fail to learn compact and clear representation clusters. With all these empirical results taken into consideration, two aspects can be summarized here: (1) there does exist a gap between the prototype and image instance embeddings extracted from the frozen backbone; (2) the shared representation transformation tends to shrink the gap between the learned prototype and image instance representations and may potentially result in the failure of learning clear and compact representation clusters for classes in the task. Our further investigation into the URL framework in Section 3.3 indicates that applying the same transformation, on the one hand, removes the discriminative information in gradients. On the other hand, our empirical results further reveal that applying the shared transformation constrains learning representations where the gap between prototypes and image instances is preserved.

Previous work [32] has demonstrated that contrastive loss helps preserve the modality gap and further contributes to improving the generalization performance of downstream tasks. Thus, in this paper, we propose a simple yet effective method, *contrastive prototype-image adaptation* (CoPA), to adapt different transformations for prototype and image embeddings similarly to CLIP [42] by treating prototypes as text prompts. In this way, the discriminative information in gradients can be preserved in different parameters, and the optimal transformations, where the gap between prototype and image representations is held, can be explored in the function space constructed by the two transformations.

Extensive experiments on the representative Meta-Dataset benchmark [49] under several task settings demonstrate that CoPA can achieve the *state-of-the-art* performance with less time consumption and learn more clear and compact representation clusters. In addition, further analysis reveals that the global minimum of the validation loss is achieved at the enlarged gap, which implies the enlargement of the gap is a process of exploring representation distributions for better generalization performance.

**Our Contribution.** In this paper, our contributions can be summarized in the following 4 parts:

- We validate there *naturally* exists a gap, which resembles the modality gap, between prototype and image instance embeddings extracted from a frozen backbone in Section 3.2.
- Our analyses in Section 3.2 reveal that the shared representation transformation shrinks the gap between prototype and image instance representations. Our investigation into the URL in Section 3.3 indicates that applying the same transformation removes the discriminative information in gradients and constrains learning representations where the gap is preserved.
- To solve this problem, in Section 4, we propose a simple yet effective method, *contrastive prototype-image adapation* (CoPA), to adapt two different transformations for prototypes and image instances as done in CLIP via substituting text prompts with prototypes.
- Extensive results reveal that CoPA can achieve the *state-of-the-art* performance on Meta-Dataset [49] (Section 5.1), enlarge the gap to achieve the global minimum of the validation loss (Section 5.2), and learn more compact image representation clusters (Section 5.2).

## 2   Preliminary

**Few-shot Task Generation.**   Consider a *meta* dataset $\mathcal{S} = \{\mathcal{S}_i\}_{i=1}^n$, where $n$ is the number of sub-datasets in $\mathcal{S}$. The sub-datasets in meta dataset satisfy that $\mathcal{S}_i \cap \mathcal{S}_j = \varnothing$, for $\forall \mathcal{S}_i, \mathcal{S}_j \in \mathcal{S}$.

Each sub-dataset $\mathcal{S}_i$ is split into three *disjoint* subsets, which are training set $\mathcal{D}_i^{\mathrm{tr}}$, validation set $\mathcal{D}_i^{\mathrm{val}}$ and test set $\mathcal{D}_i^{\mathrm{test}}$. Consistent with typical supervised learning paradigm, the meta dataset $\mathcal{S}$ is divided into two disjoint parts respectively for model training and evaluation. The datasets, in which training sets are used for pre-training, are called *seen* domains while the remaining datasets, which are only available for evaluation, are called *unseen* domains. Thus, the meta dataset can be alternatively expressed as $\mathcal{S} = \mathcal{S}^{\mathrm{seen}} \cup \mathcal{S}^{\mathrm{unseen}}$, $\mathcal{S}^{\mathrm{seen}} \cap \mathcal{S}^{\mathrm{unseen}} = \varnothing$, where the notations $\mathcal{S}^{\mathrm{seen}}$ and $\mathcal{S}^{\mathrm{unseen}}$ respectively denote the seen and the unseen domains.

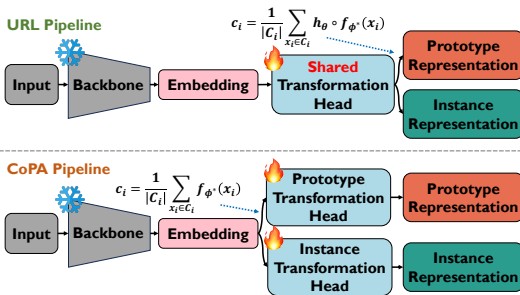

Figure 2: The **upper** subfigure shows the URL pipeline which applies the same transformation to both prototype and image instance embeddings. The **bottom** subfigure shows the pipeline of CoPA, which tries to adapt two different representation transformation heads respectively for prototypes and image instances in the way of CLIP via substituting text prompts with prototype embeddings.

Typically, a cross-domain few-shot classification task with *vary-way vary-shot* is randomly sampled in an episodic way. To be specific, at the beginning of each learning episode, a new task $\mathcal{T} = \{\mathcal{D}_\mathcal{T}, \mathcal{Q}_\mathcal{T}\}$ is sampled from a specific sub-dataset of the meta dataset $\mathcal{S}_i \in \mathcal{S}$, where $\mathcal{D}_\mathcal{T} = \{(\boldsymbol{x}_i^{\mathrm{s}}, y_i^{\mathrm{s}})\}_{i=1}^{|\mathcal{D}_\mathcal{T}|}$ denotes the *support set* of the sampled task, which is used for model training and $\mathcal{Q}_\mathcal{T} = \{(\boldsymbol{x}_i^{\mathrm{q}}, y_i^{\mathrm{q}})\}_{i=1}^{|\mathcal{Q}_\mathcal{T}|}$ denotes the *query set* of the task that is used for model evaluation.

**Problem Formulation.**   In this paper, we follow the pre-training framework adopted in previous works [12, 34, 29, 30] and build our method upon a frozen pre-trained backbone.

Consider a *frozen* pre-trained backbone $f_{\phi^*} : \mathbb{R}^{d_{\mathrm{in}}} \to \mathbb{R}^{d_{\mathrm{out}}}$ that is parameterized with a set of optimal parameters $\phi^*$ and a trainable fine-tuning module $h_\theta : \mathbb{R}^{d_{\mathrm{out}}} \to \mathbb{R}^{d_{\mathrm{out}}}$ that is parameterized with $\theta$. Given a set of support data $\mathcal{D}_\mathcal{T} = \{(\boldsymbol{x}_i, y_i)\}_{i=1}^{|\mathcal{D}_\mathcal{T}|}$, we then can obtain a set of representations $\mathcal{Z} = \{(\boldsymbol{z}_i, y_i)\}_{i=1}^{|\mathcal{D}_\mathcal{T}|}$, where $\boldsymbol{z}_i = h_\theta(f_{\phi^*}(\boldsymbol{x}_i))$, from the frozen pre-trained backbone and the fine-tuning module. The ultimate goal of the pre-training framework is learning a set of optimal task-specific parameters $\theta^*$ from the given support data so that better generalization performance can be achieved on the query data in the task. To be specific, following previous works [34, 29], the learning problem is formulated as minimizing a *nearest centroid classifier* (NCC) loss [47]:

$$\mathcal{L}(\theta) = -\frac{1}{|\mathcal{D}_\mathcal{T}|} \sum_{i=1}^{|\mathcal{D}_\mathcal{T}|} \log(p(y_i = c|\boldsymbol{z}_i; \theta)). \tag{1}$$

where $p(y = c|\boldsymbol{z}; \theta) = \frac{e^{d(\boldsymbol{z}, \boldsymbol{c}_c)}}{\sum_{c'} e^{d(\boldsymbol{z}, \boldsymbol{c}')}}$ denotes the likelihood of a sample $\boldsymbol{z}$ belonging to its class $c$, $d(\cdot, \cdot)$ denotes a measure, such as negative Euclidean distance function or cosine similarity function, to describe the similarity between samples and prototypes. In this paper, we follow URL [29] and adopt cosine similarity as the measure. Moreover, $\boldsymbol{c}_c$ denotes the prototype of class $c$, which is generated with all available samples in the class $c$ and formulated as: $\boldsymbol{c}_c = \frac{1}{|\mathcal{C}_c|} \sum_{\boldsymbol{z} \in \mathcal{C}_c} \boldsymbol{z}, \mathcal{C}_c = \{\boldsymbol{z}_i | y_i = c\}$.

## 3   Revisit the Previous Adaptation Strategy

In this section, we first revisit the adaptation strategy applied in the previous work and uncover an implicitly adopted assumption that the prototype and image instance representations are transformed with the same (linear) transformation. However, prototypes intuitively depict different levels of information from image instances. Then, we empirically demonstrate that there naturally exists a gap between prototype and image instance embeddings. Our further investigation into URL reveals that applying the same transformation to prototypes and images removes the discriminative information in gradients and constrains learning compact representation clusters where the gap is preserved.

## 3.1 An Assumption of Previous Adaptation Strategy

In previous works [29, 30] that are based on prototypes, cross-domain few-shot classification is typically formulated as a maximum likelihood estimation problem as mentioned in Eq. (1). The ultimate goal of this problem is to learn a set of representations such that classification can be performed by measuring the similarity between representations of image instances and prototypes.

Typically, the prototype of each class is calculated with all instances within the class (see Section 2 for concrete details). Assume that the transformation is defined as a linear transformation parameterized with $\Theta \in \mathbb{R}^{d_{out} \times d_{out}}$, then the calculation of the prototype $c$ of the class set $\mathcal{C}$ can be formulated as:

$$c = \frac{1}{|\mathcal{C}|} \sum_{x \in \mathcal{C}} \underbrace{(f_{\phi^*}(x)\Theta)}_{\text{Instance Representations}} = \underbrace{\left( \frac{1}{|\mathcal{C}|} \sum_{x \in \mathcal{C}} f_{\phi^*}(x) \right)}_{\text{Prototype Embeddings}} \Theta. \tag{2}$$

Since the representation transformation is usually linear, the average of image instance representations (the second part) is equivalent to the representation transformation of prototype embeddings (the third part). Thus, it is easy to observe that an assumption is implicitly adopted in the URL framework:

> *Instance-level and prototype-level embeddings share the same representation transformation.*

Intuitively, compared with image instance representations, the prototype representations contain higher-level information which is commonly shared among the instances within the class yet discriminative from that of other classes. Thus, there seems to exist a gap between prototypes and images. Such an intuition motivates us to take a further step to explore whether such a gap exists and what effect the shared representation transformation imposes on such a gap. To this end, we perform a series of analyses on the prototype and image instance embeddings and their corresponding transformed representations. In addition, we also investigate the mechanism of the adaptation strategy.

## 3.2 Empirical Analysis on Prototype and Image Instance

In this section, we conduct an analysis on the prototype and image instance embeddings and their corresponding representations obtained from the same representation transformation. Specifically, the embeddings are extracted from a frozen pre-trained ResNet-18 [18] backbone, and the representations are transformed from the embeddings with a shared linear transformation model as done in URL [29].

**"Modality" Gap between Prototype and Image Instance.** According to Liang et al. [32], there exist modality gaps between different modal embeddings, such as text and image embeddings extracted from visual language models (e.g. CLIP), and preserving such modality gaps facilitates improving the downstream performance. We notice that the prototypes, to some extent, play the same role as text prompts. Specifically, both text prompts and prototypes depict the common concepts shared among all images within the corresponding class yet discriminative from other classes. This observation motivates us to validate whether such a "modality" gap exists between prototypes and image instances.

In this section, an empirical analysis is performed on the validation set of ImageNet [44]. Following Liang et al. [32], we define the gap as the difference between the centroids of *normalized* prototype and image instance representations (embeddings) $\vec{\Delta} := \frac{1}{|\mathcal{D}_{\mathcal{T}}|} \sum_{i=1}^{|\mathcal{D}_{\mathcal{T}}|} z_i - \frac{1}{N_C} \sum_{j=1}^{N_C} c_j$, where $|\mathcal{D}_{\mathcal{T}}|$ denotes the size of support data representation set $\mathcal{Z} = \{z_i\}_{i=1}^{|\mathcal{D}_{\mathcal{T}}|}$ and $N_C$ denotes the number of classes. More detailed validation results on other datasets are available in Appendix B.

We visualize the distributions of prototype and image instance embeddings with UMAP [37] in Fig. 1(a). As shown in the figure, it is easy to observe that the prototype and image instance embeddings are located in two completely separate regions of the feature space, and there naturally exists a gap between prototype and image instance embeddings. Moreover, to verify the general existence of such a gap, we conduct an analysis with 600 randomly sampled tasks on each of all 8 seen domain datasets in Meta-Dataset. The results are reported in Table 3 in Appendix B. The results reveal that the gap also generally exists between prototype and instance embeddings in other datasets.

**Larger Gap Facilitates Generalization.** To further figure out the property of the prototype-image gap, we conduct the same embedding shift experiment as done in previous work [32]. Specifically,

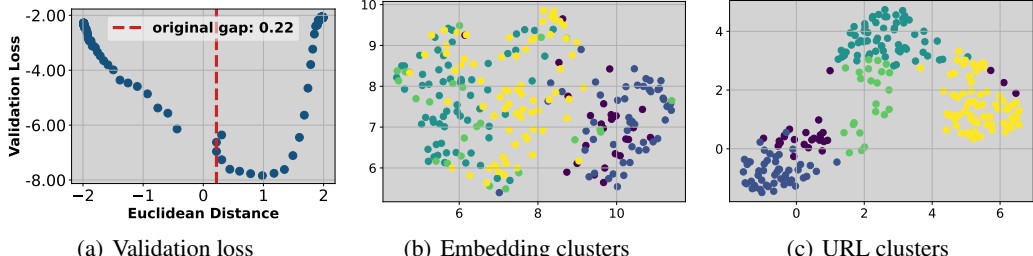

(a) Validation loss      (b) Embedding clusters      (c) URL clusters

Figure 3: **(a). The global minimum validation loss is achieved when the "modality" gap is enlarged.** Fig. (a) depicts the validation loss landscape w.r.t the changes of the "modality" gap between prototype and image instance embeddings. The validation loss fails to achieve the global minimum at the original gap, and the global minimum can be achieved when the gap is enlarged. **(b)-(c). The shared representation transformation fails to learn compact instance representation clusters.** According to the visualization results of both prototype and image instance embeddings and their representations obtained with the same representation transformation, compared to the prototype and image instance embeddings extracted from the frozen pre-trained backbone (Fig. (b)), the shared transformation fails to learn image instance representations which are well clustered (Fig. (c)).

we manually shift prototype and image instance embeddings by scaling the gap between two sets of embeddings/representations to narrow or enlarge the gap and observe the change of validation loss. The results are reported in Fig. 3(a). The X-axis depicts the size of the gap where the negative value means the position of prototype and image embeddings are mutually replaced. From the figure, it is easy to observe that the validation loss fails to achieve its global minimum at the original gap between prototype and image instance embeddings. Interestingly, we can further observe that the global minimum can be achieved by slightly enlarging the gap. This is consistent with the phenomenon observed by Liang et al. [32]. Thus, an intuitive insight is that appropriately enlarging the gap between the prototypes and image instances contributes to achieving better generalization performance.

From our perspective, the phenomenon that the larger gap facilitates improving the generalization performance can be attributed to two aspects. On the one hand, each prototype is generated with all instances within the class. Thus, data samples are naturally closer/more similar to the prototypes of their own class. This potentially results in slight overfitting. Thus, when the gap is slightly enlarged, it is equivalent to decreasing the similarity between prototypes and images. Such an enlargement, in turn, helps alleviate the overfitting and further improve the generalization performance on downstream tasks. On the other hand, as images and prototypes describe different levels of information, good generalization performance can only be achieved when the embeddings are well aligned. Thus, enlarging the gap can also be seen as exploring the optimal distributions of embeddings for alignment.

**Drawbacks of Shared Transformation.** In order to determine the effect of the shared representation transformation on representation learning, we further analyze the prototype and image instance representations obtained from the existing SOTA baseline, URL [29]. The visualization results are respectively presented in Figs. 1(b) and 3(c). According to Fig. 1(b), we can observe that the gap between prototype and image instance representations is numerically shrunken after applying the same transformation to both prototype and image instance embeddings. Meanwhile, we can also observe from Fig. 3(c) that URL fails to learn well-clustered representations for classes in the task. All these results, to some extent, demonstrate the drawbacks of applying the shared transformation.

## 3.3 Further Exploration on Shared Transformation

In this section, we conduct a further investigation to explore the effect of the shared transformation.

**Theorem 3.1.** *Let the measure $d(\cdot, \cdot)$ be the cosine similarity function. Given a set of normalized finite support data representation $\mathcal{Z} = \{(\boldsymbol{z}_i, y_i)\}_{i=1}^n$, where $||\boldsymbol{z}||_2 = 1$ for $\forall \boldsymbol{z} \in \mathcal{Z}$ and $N_C$ classes are included, then we have a lower bound of the NCC-based loss in Eq. (1):*

$$\mathcal{L}(\theta) \geq -\frac{1}{n}\sum_{i=1}^n \boldsymbol{z}_i^\top \boldsymbol{c}_c + \frac{\alpha}{n}\sum_{i=1}^n \sum_{\boldsymbol{z}' \in \mathcal{Z}} \boldsymbol{z}_i^\top \boldsymbol{z}',$$

*where $\boldsymbol{z}'$ is an independent copy of samples in $\mathcal{Z}$, $\mathcal{C}_c$ denotes sets of sample representations $\mathcal{C}_c = \{\boldsymbol{z}_i | y_i = c\}$, and $\alpha$ is a constant that satisfies $0 \leq \alpha < 1/(N_C|\mathcal{C}_j|)$ for $\forall j$.*

The complete proof is available in Appendix G.1. Theorem 3.1 provides a lower bound of Eq. (1). According to the bound, if we adopt the right side as the surrogate loss, solving Problem (1) is then equivalent to simultaneously maximizing the similarities between instances and their corresponding prototypes and minimizing the similarities among all samples. Moreover, since $z_i^\top z_j < z_i^\top z_i$ for $\forall j \neq i$, the second term can then be approximated as $\sum_{z \in \mathcal{Z}} z^\top z$ by omitting the similarity term between different image instance representations. Then, we can reformulate Problem (1) as:

$$\mathcal{L}(\Theta_\mathrm{P}, \Theta_\mathrm{I}) = -\frac{1}{|\mathcal{D}_\mathcal{T}|} \mathrm{Tr}\left(f_{\phi^*}(\boldsymbol{X})\Theta_\mathrm{I}(YY^\top f_{\phi^*}(\boldsymbol{X})\Theta_\mathrm{P})^\top\right) + \frac{\alpha}{|\mathcal{D}_\mathcal{T}|}\mathrm{Tr}\left(f_{\phi^*}(\boldsymbol{X})\Theta_\mathrm{I}\Theta_\mathrm{I}^\top f_{\phi^*}(\boldsymbol{X})^\top\right),$$

where $\boldsymbol{X} \in \mathbb{R}^{|\mathcal{D}_\mathcal{T}| \times d_\mathrm{out}}$ and $Y \in \mathbb{R}^{|\mathcal{D}_\mathcal{T}| \times N_C}$ respectively denote the support image instances and the corresponding one-hot labels, $\Theta_\mathrm{P} \in \mathbb{R}^{d_\mathrm{out} \times d_\mathrm{out}}$ and $\Theta_\mathrm{I} \in \mathbb{R}^{d_\mathrm{out} \times d_\mathrm{out}}$ denote the model parameters of linear transformation heads respectively for prototype and image instance embeddings, $\mathrm{Tr}(\cdot)$ denotes the matrix trace operation. $YY^\top f_{\phi^*}(\boldsymbol{X}) \in \mathbb{R}^{|\mathcal{D}_\mathcal{T}| \times d_\mathrm{out}}$ denotes the prototypes which are expanded to the same size of instance embeddings. In this way, the gradients w.r.t. $\Theta_\mathrm{P}$ and $\Theta_\mathrm{I}$ are:

$$\nabla_{\Theta_\mathrm{P}} \mathcal{L}(\Theta_\mathrm{P}, \Theta_\mathrm{I}) = -\frac{1}{|\mathcal{D}_\mathcal{T}|}\Theta_\mathrm{I}^\top f_{\phi^*}(\boldsymbol{X})^\top YY^\top f_{\phi^*}(\boldsymbol{X}),$$

$$\nabla_{\Theta_\mathrm{I}} \mathcal{L}(\Theta_\mathrm{P}, \Theta_\mathrm{I}) = -\frac{1}{|\mathcal{D}_\mathcal{T}|}\Theta_\mathrm{P}^\top f_{\phi^*}(\boldsymbol{X})^\top YY^\top f_{\phi^*}(\boldsymbol{X}) + \frac{2\alpha}{|\mathcal{D}_\mathcal{T}|}\Theta_\mathrm{I}^\top f_{\phi^*}(\boldsymbol{X})^\top f_{\phi^*}(\boldsymbol{X}).$$

According to the gradients above, we can observe that the gradients of $\Theta_\mathrm{P}$ depict the similarity between instance representations and prototype embeddings while the gradients of $\Theta_\mathrm{I}$ depict the similarity between prototype representations and instance embeddings. However, in the case of the shared transformation (i.e., $\Theta_\mathrm{P} = \Theta_\mathrm{I} = \Theta$), such differences are removed. Thus, the shared transformation may potentially drop the discriminative information in gradients during the adaptation phase. Meanwhile, the shared transformation also constrains the complexity of the function space, and in turn, prevents exploring the optimal representation distributions for both prototypes and images.

**Theorem 3.2 (The shared transformation).** *Consider a support data set $\mathcal{D}_\mathcal{T} = \{(\boldsymbol{x}_i, y_i)\}_{i=1}^{|\mathcal{D}_\mathcal{T}|}$ composed of $N_C$ classes and a frozen pretrained backbone $f_{\phi^*} : \mathbb{R}^{d_\mathrm{in}} \to \mathbb{R}^d$ parameterized with the optimal parameters $\phi^*$. Let $\Theta \in \mathbb{R}^{d \times d}$ be a shared linear transformation across the prototype and image instance embeddings. Then, we can obtain the image instance representations $\mathcal{Z} = \{z_i\}_{i=1}^{|\mathcal{D}_\mathcal{T}|} = \{f_{\phi^*}(\boldsymbol{x}_i)\Theta\}_{i=1}^{|\mathcal{D}_\mathcal{T}|}$, and the prototype representations $\mathcal{C} = \{\boldsymbol{c}_i\}_{i=1}^{N_C}$, where $\boldsymbol{c}_i = \frac{1}{|\mathcal{C}_i|}\sum_{z' \in \mathcal{C}_i} z' = \frac{1}{|\mathcal{C}_i|}\sum_{\boldsymbol{x}' \in \mathcal{C}_i} f_{\phi^*}(\boldsymbol{x}')\Theta$. Then we can obtain the bounds of the representation gap:*

$$m \|\Theta\|_F^2 \left\|\vec{\Delta}_\mathrm{emb}\right\|_2^2 \leq \left\|\frac{1}{|\mathcal{D}_\mathcal{T}|}\sum_{z \in \mathcal{Z}} z - \frac{1}{N_C}\sum_{\boldsymbol{c} \in \mathcal{C}} \boldsymbol{c}\right\|_2^2 \leq M \|\Theta\|_F^2 \left\|\vec{\Delta}_\mathrm{emb}\right\|_2^2,$$

*where $\vec{\Delta}_\mathrm{emb} = \frac{1}{|\mathcal{D}_\mathcal{T}|}\sum_{\boldsymbol{x} \in \mathcal{D}_\mathcal{T}} f_{\phi^*}(\boldsymbol{x}) - \frac{1}{N_C}\sum_{b=1}^{N_C}\left(\frac{1}{|\mathcal{C}_b|}\sum_{\boldsymbol{x}' \in \mathcal{C}_b} f_{\phi^*}(\boldsymbol{x}')\right)$ denotes the gap between prototype and image embeddings, $m = \min_{1 \leq i \leq d} \cos^2(\vec{\Delta}_\mathrm{emb}, \Theta^i)$ denotes the minimum value of $\cos^2(\vec{\Delta}_\mathrm{emb}, \Theta^i)$, and $M = \max_{1 \leq j \leq d} \cos^2(\vec{\Delta}_\mathrm{emb}, \Theta^j)$ denotes the maximum of $\cos^2(\vec{\Delta}_\mathrm{emb}, \Theta^j)$.*

The proof is provided in Appendix G.2.1. In Theorem 3.2, we derive the bounds of the gap between prototype and image representations learned from a shared linear transformation. For simplicity, we only consider the unnormalized representations. However, in practice, the gap is calculated with normalized representations. According to the theorem, we find that the bounds are closely related to the Frobenius norm of the transformation matrix and the angle between the embedding gap vector and the column vectors of the transformation matrix.

Theoretically, it is intractable to directly analyze the upper bound of the gap between prototype and image representations. Thus, to determine the reason for the gap shrinkage, we empirically track the change of the scale $\sqrt{M}\|\Theta\|_F$ on 600 randomly sampled tasks. The insight here is that the representation gap will not be larger than the embedding gap if the scale is smaller than 1.0. According to curves in Fig. 4, we observe that the scale is consistently smaller than 1.0 in all cases. Thus, a shared transformation cannot preserve the original gap between prototype and image instance embeddings.

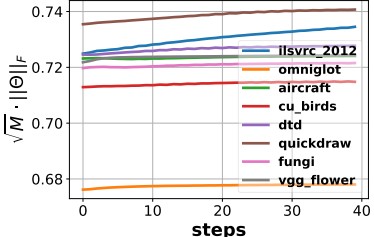

Figure 4: The change of the scale of the upper bound of URL representation gaps during the adaptation.

Table 1: **Results on Meta-Dataset under the "train on all datasets" setting.** Under the "train on all datasets" setting, the first 8 datasets are treated as "seen domians" while the last 5 are treated as "unseen domains". Mean accuracy and 95% confidence interval are reported.

| Datasets | Main Results | | | | | | | | More Learning Modules | | |
|---|---|---|---|---|---|---|---|---|---|---|---|
| | CNAPS | S-CNAPS | SUR | URT | Tri-M | FLUTE | URL | CoPA | TSA | TA²-Net | CoPA + TSA |
| **ImageNet** | 50.8±1.1 | 58.4±1.1 | 56.2±1.0 | 56.8±1.1 | **58.6±1.0** | 51.8±1.1 | 57.3±1.1 | **57.8±1.1** | 57.4±1.1 | 57.5±1.1 | **57.8±1.1** |
| **Omniglot** | 91.7±0.5 | 91.6±0.6 | 94.1±0.4 | 94.2±0.4 | 92.0±0.6 | 93.2±0.5 | 94.1±0.4 | **94.3±0.5** | **94.7±0.4** | 94.6±0.4 | 94.6±0.4 |
| **Aircraft** | 83.7±0.6 | 82.0±0.7 | 85.5±0.5 | 85.8±0.5 | 82.8±0.7 | 87.2±0.5 | 88.2±0.5 | **88.8±0.5** | 88.9±0.5 | 89.0±0.5 | **89.3±0.5** |
| **Birds** | 73.6±0.9 | 74.8±0.9 | 71.0±1.0 | 76.2±0.8 | 75.3±0.8 | 79.2±0.8 | 80.2±0.7 | **80.8±0.8** | 80.8±0.8 | 80.7±0.8 | **81.2±0.8** |
| **Textures** | 59.5±0.7 | 68.8±0.9 | 71.0±0.8 | 71.6±0.7 | 71.2±0.8 | 68.8±0.8 | 76.2±0.7 | **77.8±0.7** | 77.1±0.7 | 76.9±0.7 | **77.8±0.7** |
| **Quick Draw** | 74.7±0.8 | 76.5±0.8 | 81.8±0.6 | 82.4±0.6 | 77.3±0.7 | 79.5±0.7 | 82.2±0.6 | **82.8±0.6** | 82.2±0.6 | 82.2±0.6 | **82.7±0.6** |
| **Fungi** | 50.2±1.1 | 46.6±1.0 | 64.3±0.9 | 64.0±1.0 | 48.5±1.0 | 58.1±1.1 | 68.7±1.0 | **69.5±1.0** | 67.4±1.0 | 68.1±1.0 | **69.0±1.0** |
| **VGG Flower** | 88.9±0.5 | 90.5±0.5 | 82.9±0.8 | 87.9±0.6 | 90.5±0.5 | 91.6±0.6 | 91.9±0.5 | **92.7±0.5** | 92.5±0.5 | 92.4±0.5 | **93.0±0.5** |
| **Traffic Sign** | 56.5±1.1 | 57.2±1.0 | 51.0±1.1 | 48.2±1.1 | 63.0±1.0 | 58.4±1.1 | 63.3±1.2 | **66.6±1.1** | 83.5±0.9 | 88.3±0.8 | **88.5±0.9** |
| **MSCOCO** | 39.4±1.0 | 48.9±1.1 | 52.0±1.1 | 51.5±1.1 | 52.8±1.1 | 50.0±1.0 | 54.2±1.0 | **56.3±1.0** | 55.3±1.1 | 49.9±1.2 | **57.9±1.0** |
| **MNIST** | - | 94.6±0.4 | 94.3±0.4 | 90.6±0.5 | **96.2±0.3** | 95.6±0.5 | 94.7±0.4 | **95.2±0.4** | 96.7±0.4 | 97.0±0.4 | **97.5±0.4** |
| **CIFAR-10** | - | 74.9±0.7 | 66.5±0.9 | 67.0±0.8 | 75.4±0.8 | **78.6±0.7** | 71.9±0.8 | 73.0±0.8 | 80.3±0.8 | 76.6±0.9 | 78.7±0.8 |
| **CIFAR-100** | - | 61.3±1.1 | 56.9±1.1 | 57.3±1.0 | 62.0±1.0 | **67.1±1.0** | 62.9±1.0 | 63.4±1.0 | 70.6±1.0 | 64.5±1.2 | **70.9±0.9** |
| **Average Seen** | 71.6 | 73.7 | 75.9 | 77.4 | 76.2 | 76.2 | 79.9 | **80.6** | 80.1 | 80.2 | **80.7** |
| **Average Unseen** | - | 67.4 | 64.1 | 62.9 | 69.9 | 69.9 | 69.4 | **70.9** | 77.3 | 75.2 | **78.7** |
| **Average All** | - | 71.2 | 71.3 | 71.8 | 73.8 | 73.8 | 75.8 | **76.8** | 79.2 | 78.3 | **79.9** |
| **Average Rank** | 10.3 | 8.7 | 8.7 | 7.1 | 7.9 | 7.8 | 4.5 | **3.0** | 3.1 | 3.3 | **2.6** |

[1] For fairness, the results of URL, TSA, TA²-Net, and our proposed CoPA methods are reproduced with 5 random seeds, and we report the average of the 5 reproductions in the table. Particularly, although the reported performance of URL is lower than that in the original paper, the reproduction results are consistent with those reported on their project website. The ranks are calculated only with the first 10 datasets and only with the methods mentioned above.

Table 2: **Results on Meta-Dataset under the "train on ImageNet only" setting.** Under the "train on ImageNet only" setting, only ImageNet is treated as "seen domain" while the remaining as "unseen domains". Mean accuracy and 95% confidence interval are reported.

| Datasets | Main Results | | | | | | | | More Learning Modules | | |
|---|---|---|---|---|---|---|---|---|---|---|---|
| | Finetune | ProtoNets(large) | BOHB | FP-MAML | AFP-MAML | FLUTE | URL | CoPA | TSA | TA²-Net | CoPA+TSA |
| **ImageNet** | 45.8±1.1 | 53.7±1.1 | 51.9±1.1 | 49.5±1.1 | 52.8±1.1 | 46.9±1.1 | 57.3±1.1 | **57.7±1.1** | **57.7±1.1** | 57.4±1.1 | 57.5±1.1 |
| **Omniglot** | 60.9±1.6 | 68.5±1.3 | 67.6±1.2 | 63.4±1.3 | 61.9±1.5 | 61.6±1.4 | 69.4±1.2 | **70.9±1.2** | **73.5±1.2** | 72.8±1.2 | 73.3±1.2 |
| **Aircraft** | **68.7±1.3** | 58.0±1.0 | 54.1±0.9 | 56.0±1.0 | 63.4±1.1 | 48.5±1.0 | 57.6±1.0 | 61.6±1.0 | **65.1±1.1** | 63.5±1.0 | 64.9±1.1 |
| **Birds** | 57.3±1.3 | 74.1±0.9 | 70.7±0.9 | 68.7±1.0 | 69.8±1.1 | 47.9±1.0 | 72.9±0.9 | **74.2±0.9** | 74.0±0.9 | 73.8±0.9 | **74.7±0.9** |
| **Textures** | 69.0±0.9 | 68.8±0.8 | 68.3±0.8 | 66.5±0.8 | 70.8±0.9 | 63.8±0.8 | 75.2±0.7 | **77.0±0.7** | 76.8±0.7 | 76.6±0.7 | **77.6±0.7** |
| **Quick Draw** | 42.6±1.2 | 53.3±1.0 | 50.3±1.0 | 51.5±1.0 | 59.2±1.2 | 57.5±1.0 | 57.9±1.0 | **61.3±1.0** | 64.0±1.0 | 63.9±1.0 | **64.7±1.0** |
| **Fungi** | 38.2±1.0 | 40.7±1.2 | 41.4±1.1 | 40.0±1.1 | 41.5±1.2 | 31.8±1.0 | 46.2±1.0 | **48.0±1.1** | 46.8±1.1 | 47.6±1.1 | **48.3±1.1** |
| **VGG Flower** | 85.5±0.7 | 87.0±0.7 | 87.3±0.6 | 87.2±0.7 | 86.0±0.8 | 80.1±0.9 | 86.9±0.6 | **88.9±0.6** | 89.8±0.6 | 89.6±0.6 | **90.6±0.6** |
| **Traffic Sign** | **66.8±1.3** | 58.1±1.1 | 51.8±1.0 | 48.8±1.1 | 60.8±1.3 | 46.5±1.1 | 61.2±1.2 | 63.8±1.1 | 82.2±0.9 | **87.7±0.8** | 86.7±0.9 |
| **MSCOCO** | 34.9±1.0 | 41.7±1.1 | 48.0±1.0 | 43.7±1.1 | 48.1±1.1 | 41.4±1.0 | 53.0±1.0 | **56.1±1.0** | 55.8±1.0 | 51.3±1.2 | **57.4±1.0** |
| **MNIST** | - | - | - | - | - | 80.8±0.8 | 86.2±0.7 | **87.3±0.7** | 93.6±0.6 | 94.7±0.5 | **95.1±0.6** |
| **CIFAR-10** | - | - | - | - | - | 65.4±0.8 | 69.5±0.8 | **72.4±0.8** | **79.6±0.8** | 76.1±0.9 | 76.8±0.8 |
| **CIFAR-100** | - | - | - | - | - | 52.7±1.1 | 62.0±1.0 | **62.7±1.0** | **70.6±1.0** | 65.7±1.1 | 68.9±0.9 |
| **Average Seen** | 45.8 | 53.7 | 51.9 | 49.5 | 52.8 | 46.9 | 57.3 | **57.7** | **57.7** | 57.5 | 57.5 |
| **Average Unseen** | - | - | - | - | - | 56.5 | 66.6 | **68.7** | 72.7 | 71.9 | **73.2** |
| **Average All** | - | - | - | - | - | 55.8 | 65.9 | **67.7** | 71.6 | 70.8 | **72.0** |
| **Average Rank** | 9.3 | 7.2 | 8.0 | 9.0 | 7.1 | 10.1 | 5.3 | **4.1** | 2.5 | 3.2 | **2.2** |

[1] The results on URL, TSA, TA²-Net and our proposed methods are reproduced with 5 random seeds and reported as the average of the 5 reproduction. The ranks only consider the first 10 datasets and are calculated only with the methods in the table.

# 4 Contrastive Prototype-Image Adaptation

In this section, to address the problem mentioned above, we follow CLIP [42] and propose a simple yet effective method, _contrastive prototype-image adaptation_ (CoPA), to adapt different transformation modules respectively for the prototypes and image instances in a similar way of contrastive learning.

**Backbone Pre-training.** We pre-train a ResNet-18 [18] in the same way as URL [29] as the backbone. Specifically, 8 domain-specific backbones respectively for all seen domains are firstly pre-trained. Then, a universal encoder is distilled from these backbones and frozen during the meta-test phase.

**Contrastive Prototype-Image Adaptation.** Briefly, CoPA aims to adapt two different transformation models respectively for prototype and image instances by optimizing the symmetric cross-entropy loss. The entire pipeline of CoPA resembles that of CLIP [42]. CoPA replaces the text prompts with prototypes with the simple insight that both prototypes and text prompts depict some higher-level concepts that are common among samples within a class yet discriminative from other classes.

Specifically, consider two representation transformation heads, $h_{\theta_P}$ and $h_{\theta_I}$, respectively, for prototypes and image instances. Given a support set $\mathcal{D}_{\mathcal{T}} = \{\boldsymbol{X}, Y\}$, where $\boldsymbol{X} \in \mathbb{R}^{|\mathcal{D}_{\mathcal{T}}| \times d_{\text{out}}}$ de-

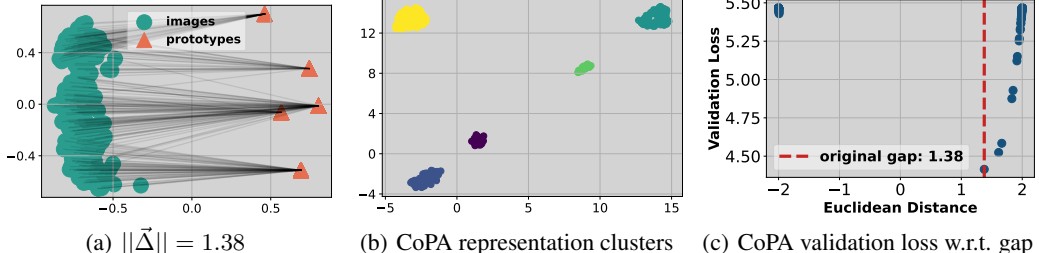

(a) $||\vec{\Delta}|| = 1.38$     (b) CoPA representation clusters     (c) CoPA validation loss w.r.t. gap

Figure 5: **(a).** The gap between prototype and image instance representations is enlarged from 0.22 to 1.38 by CoPA. Such a phenomenon is consistent with that demonstrated by Liang et al. [32]. **(b).** The clusters of image instance representations learned from CoPA. The more compact clusters reveal that CoPA learns better instance representations. **(c).** The validation loss achieves its global minimum at the gap learned by CoPA, which indicates that CoPA can improve the generalization performance.

notes the support data samples, while $Y \in \mathbb{R}^{|\mathcal{D}_\mathcal{T}| \times N_C}$ denotes the corresponding one-hot labels, where $N_C$ denotes the number of classes, we can respectively obtain the prototype representations $\boldsymbol{Z}_\mathrm{P} = h_{\theta_\mathrm{P}}(YY^\top f_{\phi^*}(\boldsymbol{X}))$ and image instance representations $\boldsymbol{Z}_\mathrm{I} = h_{\theta_\mathrm{I}}(f_{\phi^*}(\boldsymbol{X}))$. Then, CoPA learns the representations by minimizing the following symmetry cross entropy objective:

$$\min_{\theta_\mathrm{P},\theta_\mathrm{I}} \mathcal{L}(\theta_\mathrm{P},\theta_\mathrm{I}) := \mathcal{L}_\mathrm{CE}(\frac{1}{\tau}\boldsymbol{Z}_\mathrm{I}\boldsymbol{Z}_\mathrm{P}^\top, Y_\mathrm{pseudo}) + \mathcal{L}_\mathrm{CE}(\frac{1}{\tau}\boldsymbol{Z}_\mathrm{P}\boldsymbol{Z}_\mathrm{I}^\top, Y_\mathrm{pseudo}), \tag{3}$$

where $\tau$ is a temperature coefficient, $Y_\mathrm{pseudo} = [0, 1, ..., |\mathcal{D}_\mathcal{T}| - 1]$ denotes the pseudo labels, and $\mathcal{L}_\mathrm{CE}$ denotes the cross-entropy loss. The complete CoPA algorithm is presented in Algorithm 1.

**Discussion.** Two aspects are worth noticing in our proposed CoPA. On the one hand, two different transformations are adopted in our proposed CoPA method. In this way, the discriminative information in gradients is preserved in different sets of parameters. Besides, introducing different transformations expands the function space, which further facilitates learning better prototype and image instance representations. On the other hand, the contrastive learning objective, which has been demonstrated to facilitate preserving the modality gap, is adopted as the learning objective. In particular, to match this objective, we modify the prototype embeddings from $Y^\top f_{\phi^*}(\boldsymbol{X})$ to $YY^\top f_{\phi^*}(\boldsymbol{X})$ so that the size of the prototype embeddings is consistent with that of image instance embeddings. Such a modification further indicates the cluster structure of the given support data set. Thus, by minimizing the symmetric cross entropy loss, the two transformations are encouraged to align the prototype and image instance representations and learn more class-specific and discriminative representations.

## 5 Experiments

In this section, we evaluate CoPA on Meta-Dataset [49] under both "train on all datasets" and "train on ImageNet only" settings with a series of tasks to answer the following questions: (1) Does CoPA achieve better generalization performance? (2) Does CoPA benefit from extra learning modules? (3) Why does CoPA perform better? Detailed experimental settings are available in Appendix E.

### 5.1 Main Results & Analysis

In this section, we evaluate the generalization performance of our proposed CoPA method on Meta-Dataset under both "train on all datasets" and "train on ImageNet only" settings. In order to demonstrate the effectiveness of CoPA, we compare CoPA against several currently *state-of-the-art* baselines, including fo-Proto-MAML (FP-MAML) [49], ALFA+fo-Proto-MAML (AFP-MAML) [2], CNAPs [43], Simple-CNAPs [3], SUR [12], URT [34], Tri-M [35], FLUTE [50], URL [29].

**Train on All Datasets.** We report the evaluation results under the "train on all datasets" setting in Table 1. As shown in the table, it is easy to observe that CoPA achieves the best performance on 9 out of 13 datasets among all approaches and outperforms URL in all cases. Specifically, compared with URL, our proposed CoPA method achieves 0.7%, 1.5%, and 1.0% improvements on average respectively on seen, unseen, and all domains. It is also worthwhile to notice that CoPA performs better on unseen domains. Specifically, our proposed CoPA method achieves 3.3%, 2.1%, and 1.1% improvements respectively on Traffic Sign, MSCOCO, and CIFAR 10 datasets. Meanwhile, we also

notice that CoPA achieves comparable or even better results on seen domains compared with TSA [30] which plugs extra learning modules into the frozen backbone for more task-specific parameters.

**Train on ImageNet Only.** The results under the "train on ImageNet only" setting are reported in Table 2. Briefly, CoPA achieves the best performance on 11 out of 13 datasets and consistently outperforms URL in all cases. On average, CoPA achieves $0.4\%$, $2.1\%$, and $1.8\%$ improvements respectively on seen, unseen, and all domains. Moreover, the results also reveal that CoPA consistently performs better on unseen domains. Specifically, CoPA achieve impressive improvements on Omniglot ($1.5\%$), Aircraft ($4\%$), Birds ($1.3\%$), Textures ($1.8\%$), Quick Draw ($3.4\%$), Fungi ($1.8\%$), VGG_Flower ($2\%$), Traffic Sign ($2.6\%$), MSCOCO ($3.1\%$), MNIST ($1.1\%$), CIFAR ($2.9\%$ & $0.7\%$).

**Extra Learning Modules.** We follow TSA [30] to plug extra trainable modules into the frozen backbone.The results are reported respectively in Table 1 and 2. According to the table, we can observe that CoPA benefits from these extra modules and achieves better performance than previous state-of-the-art approaches (e.g., TSA and TA$^2$-Net). CoPA + TSA outperforms 11 out of 13 datasets under the "train on all datasets" setting. Similarly, CoPA + TSA consistently achieves better generalization performance on unseen domains.Specifically, CoPA + TSA achieves $5\%$, $2.6\%$, and $0.8\%$ improvements respectively on Traffic Sign, MSCOCO, and MNIST datasets. Under the "train on ImageNet only" setting, CoPA + TSA achieves the best performance on 7 out of 13 datasets. Compared with TSA, CoPA + TSA achieves better performance on Birds ($0.7\%$), Textures ($0.8\%$), Fungi ($1.5\%$), VGG_Flower ($0.8\%$), Traffic Sign ($4.5\%$), MSCOCO ($1.6\%$) and MNIST ($1.5\%$).

## 5.2 Discussion: Why CoPA Performs Better?

In previous experiments, we have demonstrated that CoPA can achieve better generalization performance. In this section, we aim to validate whether CoPA preserves the gap and learns better representations. Further, we would like to explore why CoPA performs better than previous works.

**More Results.** To verify the effectiveness of CoPA, we conduct more analyses. Firstly, we investigate the representation shift experiment to observe the gap between prototype and image representations. As shown in 5(a), in contrast to URL, the gap between prototype and image representations learned via CoPA is enlarged. Such a phenomenon demonstrates that CoPA can *hold* the gap between prototypes and image instances. In addition, we further check the image instance representations learned with CoPA. According to Fig. 5(b), the image representations are more clearly and compactly clustered than Fig. 3(c). This case reveals that CoPA can facilitate learning more class-specific representations.

### 5.2.1 Ablation Study

The results reported above indicate that CoPA can enlarge the representation gap and learn well-clustered image representations simultaneously. In order to further figure out why CoPA is able to improve performance, in this section, we propose to conduct ablation studies on the two important components of CoPA: the different transformations and SCE loss. The results are reported in Table 10.

**Ablation: Different Transformation Modules.** As we have discussed in Section 3.3, the shared transformation may drop the discriminative information in gradients. Thus, we propose to substitute the shared transformation with the different transformations adopted in CoPA. As shown in the table, although URL+2Heads achieves slightly better results in some cases, its performance drops in the remaining cases. By plotting the test loss curves (cf. Fig. 6), we can observe that overfitting takes place.

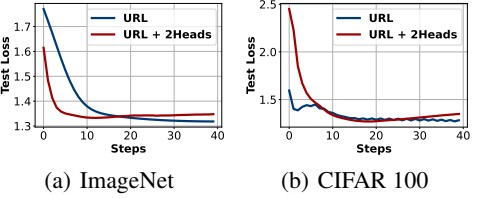

(a) ImageNet      (b) CIFAR 100

Figure 6: Comparison of test losses of URL and URL+2Heads on ImageNet and CIFAR100.

**Ablation: SCE Loss.** From the table, we can observe that URL with SCE loss achieves comparable performance to the URL baseline in most datasets. Meanwhile, URL with SCE achieves significantly better results on some datasets, such as Fungi and MSCOCO, in which URL baseline tends to overfit (cf. Figs. 27(g) and 27(j)). This indicates that SCE loss can improve generalization performance. In addition, we also replace the SCE loss with the NCC loss on CoPA. According to the results of "CoPA + NCC" shown in the table, it is easy to find that the performance in all cases drops drastically. For example, the performance on Fungi and MSCOCO respectively decreases by $11.6\%$ and $18.7\%$. All these empirical results demonstrate that SCE loss facilitates improving generalization performance.

### 5.2.2 Further Discussion

**Effectiveness of Different Transformations.** As aforementioned, an advantage of CoPA is applying two different transformations respectively to prototype and image instance embeddings. The goal of applying two different transformations in CoPA is to preserve the discriminative information of gradients in different sets of parameters. Meanwhile, applying two different transformations also increases the complexity of the hypothesis space, which contributes to reducing the approximation error and learning representations in a more flexible way. However, simply increasing hypothesis space complexity may also result in overfitting (cf. Fig. 6). According to Fig. 27, CoPA, which is equipped with SCE loss, benefits from the increased complexity and achieves better performance.

**Essence of Gap Enlargement.** In Fig. 3(a), we observe that the global minimum of the validation loss can be achieved by slightly enlarging the gap between prototype and images. Meanwhile, according to Fig. 5(c), the validation loss achieves its global minimum at the enlarged gap learned with CoPA. As aforementioned, we conjecture that the reasons for the phenomenon that gap enlargement facilitates generalization mainly include two aspects: representation alignment and overfitting alleviation.

We first start with representation alignment. Typically, representation alignment is performed with SCE loss. Specifically, we omit the temperature coefficient $\tau$ and simply rewrite Eq. (3) as:

$$\mathcal{L}_{\text{SCE}} = -\frac{1}{|\mathcal{D}_\mathcal{T}|} \sum_{i=1}^{|\mathcal{D}_\mathcal{T}|} \log \frac{\exp(\boldsymbol{z}_i^\top \boldsymbol{c}_i)}{\sum_{j=1}^{|\mathcal{D}_\mathcal{T}|} \exp(\boldsymbol{z}_i^\top \boldsymbol{c}_j)} - \frac{1}{|\mathcal{D}_\mathcal{T}|} \sum_{i=1}^{|\mathcal{D}_\mathcal{T}|} \log \frac{\exp(\boldsymbol{c}_i^\top \boldsymbol{z}_i)}{\sum_{j=1}^{|\mathcal{D}_\mathcal{T}|} \exp(\boldsymbol{c}_i^\top \boldsymbol{z}_j)}. \quad (4)$$

**Theorem 5.1.** *Given a set of normalized finite support data representation $\mathcal{Z} = \{(\boldsymbol{z}_i, y_i)\}_{i=1}^n$ and a set of normalized prototype representations $\mathcal{C} = \{\boldsymbol{c}_i\}_{i=1}^n$, where $||\boldsymbol{z}||_2 = 1$ for $\forall \boldsymbol{z} \in \mathcal{Z}$ and $||\boldsymbol{c}||_2 = 1$ for $\forall \boldsymbol{c} \in \mathcal{C}$, then we are able to obtain a lower bound of SCE loss in Eq. (4):*

$$\mathcal{L}_{\text{SCE}} \geq -\frac{2}{n} \sum_{i=1}^n \boldsymbol{z}_i^\top \boldsymbol{c}_i + \frac{2}{n} \sum_{i=1}^n \sum_{k=1}^{N_C} \frac{|\mathcal{C}_k|}{n} \boldsymbol{z}_i^\top \boldsymbol{c}_k,$$

*where $\mathcal{C}_k$ denotes the set of support data of the class $k$ and $N_C$ denotes the number of classes.*

The proof is in Appendix G.3. According to Theorem 5.1, we find that the lower bound of SCE loss plays a similar role to that of NCC loss. In detail, the lower bound aims at maximizing the similarity between each sample and its corresponding prototype while minimizing the similarities between the sample and all prototypes. When minimizing the lower bound as the surrogate loss, the second term, which measures the similarities between image and prototype representations, is minimized. Thus, the Euclidean distances between representations of prototypes and images (i.e., the gap) are enlarged.

An interesting point is that the similarities between images and prototypes are minimized with the weights calculated based on the size of $\mathcal{C}_k$. Differently, in NCC loss, the weights are fixed to $\frac{1}{N_C}$ for all cases. Thus, the similarities between samples and the prototypes involving more samples will be significantly reduced. Consequently, the gap between images and prototypes tends to be enlarged.

For the overfitting, on the one hand, the potential overfitting discussed in Section 3.2 is mitigated by applying different transformations. On the other hand, the high similarities between samples and prototypes may also result in overfitting. For example, in Fig. 3(a), the validation loss increases when the gap is narrowed. However, as we have discussed above, minimizing SCE loss tends to reduce such similarities. Thus, the gap enlargement also functions as a regularization to alleviate overfitting.

Therefore, with all aspects taken into consideration, the essence of CoPA is a representation alignment of prototypes and images, in which a balance between learning discriminative representations and achieving better generalization performance, is explored in a more flexible hypothesis space.

## 6 Conclusion

In this paper, we validate that there naturally exists a gap, which resembles the modality gap in visual language models, between prototype and image instance embeddings, and uncover that the shared transformation shrinks such a gap and constrains the learning of well-clustered representations. In order to solve these problems, we propose CoPA to adapt different transformations respectively for prototype and image instances via optimizing SCE loss. Empirical results on Meta-Dataset reveal that CoPA can achieve *state-of-the-art* performance. Our further analyses indicate that the essence of CoPA is a representation alignment, where a balance between learning discriminative representations and achieving better generalization performance is explored in a more flexible hypothesis space.

## Acknowledgement

HDT, ZKZ and BH were supported by Guangdong Basic and Applied Basic Research Foundation Nos. 2022A1515011652 and 2024A1515012399, NSFC General Program No. 62376235, HKBU Faculty Niche Research Areas No. RC-FNRA-IG/22-23/SCI/04, and HKBU CSD Departmental Incentive Scheme. FL is supported by the Australian Research Council (ARC) with grant numbers DP230101540 and DE240101089, and the NSF&CSIRO Responsible AI program with grant number 2303037. TLL was partially supported by the following ARC projects: FT220100318, DP220102121, LP220100527, LP220200949, and IC190100031.

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

# Appendix

## Limitations

In this paper, in order to solve the problem that the shared transformation adopted in previous works potentially hurts the discriminative information in gradients and constrains learning representations where the gap is preserved, we propose to fine-tune two different transformations respectively for prototype and image instances in the same way of CLIP by treating the prototypes as text prompts.

One potential limitation of our proposed CoPA method is the symmetric cross-entropy. Symmetric cross entropy loss is commonly applied in contrastive learning. In contrastive learning, the batch size of the data samples is an important issue for better downstream tasks. Larger batch size usually means better downstream performance. We also notice this phenomenon in this paper. Although CoPA can still achieve relatively better performance, the performance tends to degrade when the size of the data batch becomes small. Thus, this constrains the application of CoPA to some extent.

## Broader Impact

In this paper, we find that there exists a gap, which resembles the modality gap in visual language models, between prototype and instance embeddings. An intuition of such a phenomenon is that prototypes are generated with all data samples within a specific class and consequently can be treated as an abstract concept of all data samples in the class. From this perspective, the prototype information plays the same role as text prompts in visual language models to some extent. Thus, our work provides a direction that we can extract some abstract concept information from the available single modal data and then train/adapt models with such information in some appropriate way, such as contrastive learning, to learn more discriminative and useful representation for downstream tasks. Since visual information is more diverse than texts, we think it will be a promising and feasible way of developing general models for various kinds of visual tasks. Based on this, there are many potential societal consequences of our work, none of which we feel must be specifically highlighted.

## A   Related Work

**Cross-domain few-shot classification.**   Cross-domain few-shot classification is a learning paradigm that aims at learning to perform classification on previously unseen data and domains by merely fast training a model on several labeled training data. Generally, the cross-domain few-shot classification problem can be seen as an extension of conventional few-shot classification problem [52, 14, 47, 6]. The main difference between conventional and cross-domain few-shot classification lies in the domains for evaluation. In the conventional few-shot classification setting, a model is trained and evaluated on the same dataset (domain), where the distribution is invariant. However, in the cross-domain few-shot classification setting, a model is trained on the training sets of several datasets with different distributions and then evaluated on the test sets of datasets that are used for training as well as the datasets that have never been observed during the training/adaptation phase. Thus, compared with the conventional few-shot classification task, the cross-domain few-shot classification task is much more challenging since there exist distribution gaps between the training and previously unseen test datasets. Currently, all existing works regarding cross-domain few-shot classification are mainly based on conventional meta-learning methods, such as Prototypical Networks [47] and MAML [14]. The former is famous for its strong ability to learn effective inductive bias while the latter is impressive for its flexibility of adaptation to new tasks. Generally, the existing works can be divided into two different types according to their ways of training the backbones and classifiers.

Some of the existing works train the model from scratch with a sequence of few-shot tasks in the same way as conventional few-shot learning frameworks. A typical case of this kind of learning paradigm is Proto-MAML [49]. Specifically, Proto-MAML proposes to train a model with the modified Prototypical loss in the same way as MAML to take advantage of the strengths of the two frameworks. We would like to note that Prototypical loss plays an important role in current cross-domain few-shot classification research. The reason behind this is that the tasks adopted in cross-domain few-shot classification tasks are usually sampled with varied numbers of classes and shots [49]. In this case, it is infeasible to train a classifier in the same way as the conventional meta-learning framework, where a parameterized classifier is initialized with a fixed number of classes. In addition to Proto-MAML, more works are proposed to solve cross-domain few-shot classification tasks. CNAPs [43] proposes to generate parameters with feature-wise linear modulation (a.k.a., FiLM [41]) for task-specific modules

in their model. However, a potential problem of CNAPs is that the Euclidean distance adopted in the loss implicitly assumes that each cluster is distributed according to a unit normal. This may further result in misclassification of data points. Based on this observation, SimpleCNAPs [3] further proposes a non-parametric classifier from the perspective of class-covariance-based distance metric to improve the classification performance of the original CNAPs method. Moreover, Doersch et al. [10] notice that the representations learned via the Prototypical Network framework may only represent the observed data yet discard the information that might contribute to the recognization of out-of-distribution classes. Thus, in order to solve this problem, they further propose CrossTransformer to leverage query image information in pixel-level during the adaptation steps via an attention module.

Different from the learning frameworks that train the entire pipeline from scratch, the other kinds of methods usually respectively pre-train one or several domain-specific backbones on all available datasets during the training phase in a conventional supervised learning paradigm. Then, during the evaluation phase, the prior knowledge in these pre-trained backbones is transferred to the new tasks through representation fusion or representation adaptation. In cross-domain few-shot classification community, SUR [12] firstly proposes to select relevant features from the 8 pre-trained backbones to represent the data of the given new task via a vector. Later, URT [34] further proposes to quickly train a multi-head transformer on top of 8 pre-trained backbones to learn to select features for unseen data. Besides, Triantafillou et al. [50] proposes to treat the convolutional layers in a model as universal templates while treating the batch normalization layers as the domain-specific modules. Thus, FLUTE is proposed to pre-train a set of universal weights and several domain-specific BN modules and then train a small model to learn to combine these BN layers and transfer the prior knowledge to the new tasks. Although leveraging pre-trained backbones is able to take advantage of high-quality representations learned from previous domains, it is quite time-consuming to perform several forward passes during both training and evaluation phases. In order to solve this concern, Li et al. [29] proposes URL to distill the prior knowledge of all 8 pre-trained backbones in one single backbone. Then, during the test phase, a simple linear layer is fast fine-tuned on top of the distilled multi-domain backbone to transfer the priori to the unseen domains. Later, TSA [30] is proposed to enhance URL via plugging more learnable residual modules into the frozen pre-trained backbone to learn more task-specific features for unseen data. Meanwhile, recent work $TA^2$-Net [16] notices that different tasks prefer different learnable modules and proposes to train an Action Generation Network in the way of reinforcement learning to learn to select the optimal module for each task.

**Contrastive loss and contrastive learning.** The ultimate goal of contrastive learning is to learn discriminative and robust representations at the instance level by comparing two augmented counterparts of an image. In practice, the goal is realized through contrastive loss which focuses on maximizing the similarity between positive pairs while minimizing the similarity between negative pairs.

Compared with early contrastive loss [8, 17, 46], the current contrastive loss is constructed upon a batch of data, where the number of negative samples is larger than that of positive samples. Such contrastive loss was first proposed by Sohn [48] as multi-class N-pair loss and then improved and further explored by further works [5, 19, 7, 15, 55, 4]. Based on the InfoNCE, Li et al. [31] further explores an alternative loss from the perspective of the Hilbert-Schmidt Independence Criterion measure. Different from conventional unsupervised contrastive learning, Khosla et al. [23] proposes a fully supervised contrastive learning paradigm, SupCon, via leveraging label information. To be specific, in conventional unsupervised contrastive learning, only the augmented counterparts of an image are treated as positive images. However, in supervised contrastive learning, all images that share the same "general" concept are treated as positive samples. The ultimate goal of SupCon is to pull normalized embeddings from the same class closer than embeddings from different classes. Then, with the emergence of pre-trained vision language models, such as CLIP [42], embeddings that are generated from other modalities are treated as supervised information and have shown impressive power in multi-modality learning fields. From our perspective, SupCon and CLIP share the same core. The key difference is that SupCon learns representations by comparing each sample with all other samples within the same class while CLIP learns to align the representations of each image sample with the representations of text information that are more "general" than a set of images.

**Multi-modal contrastive representation learning.** Multi-modality learning [21, 28, 54, 56] is a learning paradigm that aims at aligning representations extracted from different data modalities, such as text and images. Since the emergence of CLIP [42] has shown impressive performance on downstream tasks, more and more attention has been attracted to this research topic. Recently, Liang

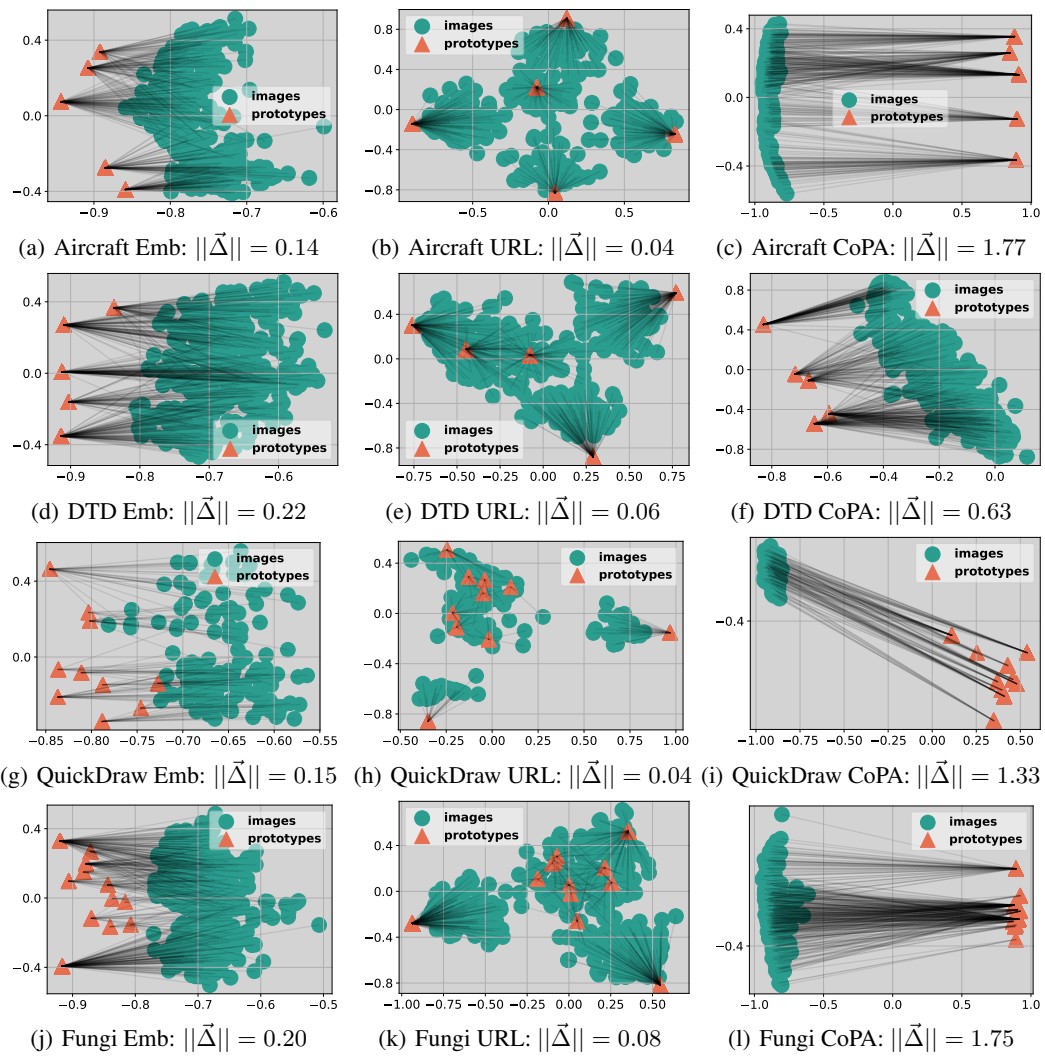

Figure 7: **"Modality" gaps between prototypes and images on some other datasets.** We selected a part of the visualization results regarding the "modality" gap between embeddings and learned representations of prototypes and image instances. It is easy to observe that URL shrinks the gap while CoPA enlarges the gap between the prototype and image instance representations.

et al. [32] argues that there exists a modality gap between different modal data and theoretically demonstrates this from the perspective of cone effect [38, 13]. Further, Liang et al. [32] demonstrates that such a modality gap contributes to the downstream performance on zero-shot learning tasks. In this work, following Liang et al. [32], we find that such kind of "modality" gap also exists between prototype and instance embeddings in the context of single modality data, and applying the same transformation to both of these two kind of embeddings shrinks such a gap. In order to solve this problem, we propose CoPA to adapt two different transformation heads respectively for prototypes and instances in the way of CLIP where text prompts are substituted with prototype embeddings.

## B    Detailed Study on Gaps between Prototypes and Images

As we have mentioned in the previous section, a prototype is typically calculated with all available data samples (images) within the class. Thus, compared to image instances, the prototypes contain representative representations, which are commonly shared among image instances within the class. From this perspective, we notice that the prototypes play a similar role to the text prompts adopted in visual language models (e.g., CLIP [42]). The text prompts describe the general and representative

Table 3: **Average gaps between prototype and image instance embeddings/representations.** The results are the average value of the gaps of 600 random tasks under "train on all datasets" settings.

| Gaps | ImageNet | Omniglot | Aircraft | Birds | Textures | QuickDraw | Fungi | VGG_Flower |
|---|---|---|---|---|---|---|---|---|
| $\lVert\bar{\Delta}_{\mathrm{embed}}\rVert$ | 0.19 | 0.11 | 0.12 | 0.12 | 0.14 | 0.14 | 0.14 | 0.14 |
| $\lVert\bar{\Delta}_{\mathrm{URL}}\rVert$ | 0.17 | 0.09 | 0.08 | 0.07 | 0.07 | 0.06 | 0.06 | 0.06 |
| $\lVert\bar{\Delta}_{\mathrm{CoPA}}\rVert$ | 0.60 | 0.42 | 0.55 | 0.54 | 0.54 | 0.54 | 0.56 | 0.59 |

information of a category. Thus, we conjecture that the prior information respectively contained in prototypes and image instances is at different levels. To be specific, image instance representations are encoded with instance-level information closely related to the contents in the given image while prototype representations are a set of class-specific representations. According to the visualizations in [32], the image and text prompt embeddings extracted from visual language models are located in completely separated areas of the feature space. In that work, the reason for such a phenomenon is attributed to the different kinds of modalities of the input data. Thus, the gap between the clusters of the extracted text and image embeddings is correspondingly called the modality gap.

As aforementioned, the prototype resembles the text prompt to some extent since both of them describe some high-level information commonly shared among images within a class. Thus, similar to the modality gap, there might also exist some type of gap between prototype and image representations. To further determine our intuition, in this section, we propose to conduct an analysis to figure out whether the gap between the prototype and image instance embeddings/representations exists.

**Analysis Settings.** To examine the gap between prototypes and image instances, we follow Liang et al. [32] to measure the Euclidean distance between the prototypes and image instances embeddings/representations. To be specific, in the analysis, tasks are first randomly sampled from the validation set of a specific dataset, such as ImageNet [44], and the corresponding embeddings are obtained by feeding support data of the tasks into a frozen pre-trained ResNet-18 backbone. Then, the prototype for each class in the task is calculated as a mean of all support data within the class following previous works [47, 49, 43, 3, 12, 34, 29, 30]. By respectively applying URL [29] and our proposed CoPA methods on the extracted embeddings, the corresponding representations are obtained. Finally, we reduce the dimension of the prototype and image representations via some dimension reduction tools, such as UMAP [37], and visualize the results. The results are visualized in Figs. 1, 5(a), 7, and 19. According to the visualization results, there are several interesting phenomena:

- According to Figs. 1(a), 7(a), 7(d), 7(g), 7(j), and the results of other datasets (Fig. 19), it is easy to observe that the prototype and image instance embeddings extracted from a frozen pre-trained backbone tend to be located in different regions of the feature space. This phenomenon is consistent with what happens in visual language models [32]. It also demonstrates the aforementioned intuition that prototype and image instance data describe the information at different levels and should be treated differently during the adaptation.

- Then, according to Figs. 1(b), 7(b), 7(e) and 7(k), we can observe that the "modality" gap is shrunken after applying URL, which imposes an assumption that the same representation transformation is applied to both prototype and image instance embeddings. This phenomenon demonstrates that sharing the same transformation between prototypes and image instances potentially damages the distribution differences between the prototypes and image instances and in turn removes the differences of semantic information between them.

- Finally, according to Figs. 5(a), 7(c), 7(f) and 7(l), we can observe that the representations learned from our proposed CoPA method preserve the "modality" gap and the gap is further enlarged compared with the gap between the embeddings. As claimed in Liang et al. [32], the enlargement of the gap contributes to the improvements in generalization performance. According to our empirical results in Tables 1 and 2, CoPA achieves the SOTA performance and significantly outperforms URL and URL + TSA, which is consistent with the conclusion. Thus, we can further say that it is the gap between prototype and image instance representations that facilitates CoPA to achieve better empirical performance.

In addition to these visualization results, we take a further step to validate whether this phenomenon is common among all tasks of all seen domain datasets. To this end, we collect the results of the gaps of the extracted embeddings and the representations respectively learned from URL and CoPA among 600 randomly sampled tasks under the "train on all datasets" task setting with the

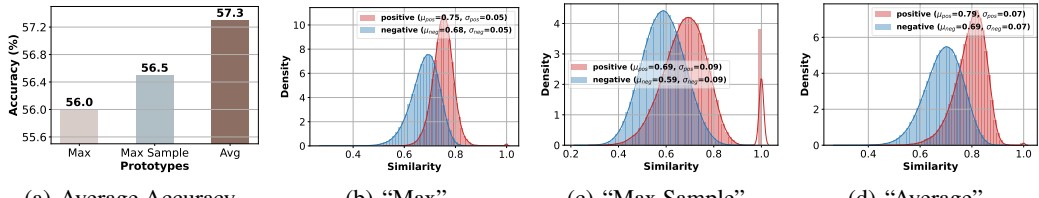

| (a) Average Accuracy | (b) "Max" | (c) "Max Sample" | (d) "Average" |

Figure 8: **Analysis results of three kinds of prototypes: "Max", "Max Sample" and "Average" prototypes on ImageNet dataset. (a).** Average accuracy of URL with three kinds of prototypes on ImageNet dataset. The results are obtained by averaging 5 reproductions with different random seeds (i.e. 41-45). **(b).** The distribution of both positive and negative similarities with the "Max" prototype. **(c).** The distribution of both positive and negative similarities with the "Max Sample" prototype. **(d).** The distribution of both positive and negative similarities with the "Average" prototype.

random seed 42 on the validation sets of all seen domain datasets. The results are reported in Table 3. The gaps are calculated following Liang et al. [32]. Given a set of normalized support data embeddings/representations $\{z_i\}_{i=1}^{|\mathcal{D}_\mathcal{T}|}$ and a set of normalized prototypes generated from the support data $\{c_j\}_{j=1}^{N_C}$, the gap is calculated as the difference between the averaged prototypes and images:

$$\vec{\Delta} := \frac{1}{|\mathcal{D}_\mathcal{T}|} \sum_{i=1}^{|\mathcal{D}_\mathcal{T}|} z_i - \frac{1}{N_C} \sum_{j=1}^{N_C} c_j.$$

Thus, in this way, with the normalized prototype and image instance embeddings/representations, the "modality" gap is further defined as the Euclidean distance between the two sets of embeddings/representations $||\vec{\Delta}||$, where $|| \cdot ||$ denotes the Euclidean distance.

According to the results reported in Table 3, it is easy to observe that URL consistently shrinks the gaps between prototypes and image instances over all datasets while CoPA consistently enlarges the gaps. These results indicate that URL undermining the "modality" gaps while CoPA preserving and enlarging the gaps is not an occasional case. It commonly happens across all domains.

## C   Detailed Study on Prototype Selection

In this paper, in order to select the most appropriate prototypes for CoPA, we conduct an analysis to perform prototype selection. Generally, the prototype for each class ought to contain the most general and discriminative information regarding the class. To this end, three kinds of prototypes are selected as candidates. We will provide detailed descriptions and analyses of them in the following parts.

### C.1   Introduction of Three Prototypes

**Max Prototype.** The "Max" prototype is defined as the sample that is composed of the largest features in each dimension among all support data embeddings extracted from a frozen backbone. The pseudo codes are provided in the following:

```python
import torch

def max_prototype(X:torch.tensor, Y:torch.tensor):
    '''
    Args:
        X: shape: [n_samples, n_dims], the support data embeddings;
        Y: shape: [n_samples,], the support data labels.
    Return:
        max_protos: torch.tensor, shape: [n_classes, n_dims].
    '''
    n_way = len(Y.unique())
    max_protos = torch.zeros(n_way, X.shape[-1]).type_as(X.dtype)
    for i in range(n_way):
        max_proto[i], _ = X[(Y == i), :].max(dim=0)
    return max_protos
```

**Max Sample Prototype.** The "Max Sample" prototype is the data sample that has the highest similarities over all other samples within its class set. The pseudo codes are provided in the following:

```python
import torch

def max_sample_prototype(X:torch.tensor, Y:torch.tensor):
    '''
    Args:
        X: shape: [n_samples, n_dims], the support data embeddings;
        Y: shape: [n_samples,], the support data labels.
    Return:
        maxsample_protos: torch.tensor, shape: [n_classes, n_dims].
    '''
    n_way = len(Y.unique())
    maxsample_protos = torch.zeros(n_way, X.shape[-1]).type_as(X.dtype
                                          )
    for i in range(n_way):
        class_samples = X[(Y == i), :]
        num_samples = class_samples.shape[0]
        normed_samples = class_samples / class_samples.norm(dim=1,
                                            keepdim=1)
        similarity_mat = normed_samples @ normed_samples.t()
        max_proto[i] = class_samples[torch.argmax(similarity_mat.sum(1
                                            ), :]
    return maxsample_protos
```

**Average Prototype.** The "Average" prototype is the typical prototype adopted in previous few-shot classification works and calculated as the average vector of all support data embeddings extracted from a frozen pre-trained backbone within the given class. The pseudo codes are as the following:

```python
import torch
import torch.nn.functional as F

def avg_prototype(X:torch.tensor, Y:torch.tensor):
    '''
    Args:
        X: shape: [n_samples, n_dims], the support data embeddings;
        Y: shape: [n_samples,], the support data labels.
    Return:
        avg_protos: torch.tensor, shape: [n_classes, n_dims].
    '''
    one_hot_labels = F.one_hot(Y)     # shape: [n_samples, n_classes]
    avg_protos = Y.t() @ X / Y.t().sum(dim=1, keepdim=True)

    return avg_protos
```

### C.2   Analysis Settings & Results

**Settings.** The analysis regarding prototype selection is performed on the support data embeddings (of tasks sampled from the validation set of seen domain datasets) extracted from the frozen pre-trained backbone. The results are obtained from 600 randomly sampled with the random seed 42. Specifically, for each sampled task, the support data are fed into the backbone for the corresponding embeddings. Then, the prototype embeddings for each class are calculated respectively. With the support data and prototype embeddings, we measure two kinds of similarity. The first one is the similarity between each support data sample and the prototype that it belongs to. We call such kind of similarity "positive similarity". The other one is the similarity between each sample and the prototypes of other classes. We call such kind of similarity "negative similarity". The density of both kinds of similarity of the three prototypes (over all seen domain datasets) are visualized in Fig. 8 and Figs. 12 - 18.

**Results.** As aforementioned, the prototype should contain information that is commonly shared among samples within the class while distinctive from other classes. In other words, such a prototype has high similarity with the samples within the class while low similarity with the samples from

other classes. Thus, for prototypes that depict the representative features of the classes, the positive similarity should be as large as possible while the negative similarity should be as small as possible.

According to the results on ImageNet dataset shown in Fig. 8, the comparison of the three kinds of prototypes mainly includes two parts. On the one hand, the numerical results of the few-shot classification task are evaluated respectively with the three prototypes. On the other hand, the distributions of "positive similarity" and "negative similarity" are respectively measured.

From the perspective of numerical results, it is easy to observe that the "Average" prototype achieves the best empirical generalization performance on ImageNet and outperforms the "Max" and the "Max Sample" prototypes. In addition, the "Average" prototype also achieves the best results in positive similarity distribution with an average positive similarity of 0.79 (see Fig. 8(d)). Interestingly, the "Max Sample" prototype, which owns the highest similarity over all other samples, obtains the lowest positive similarity, even smaller than the "Max" prototype. We conjecture the reason for this phenomenon is that the "Max Sample" prototype does not contain the common features shared in its class though it owns the highest similarity among all other samples within its class. Thus, with all aspects taken into consideration, we select the "Average" prototype for our proposed CoPA.

## D    Detailed Task Settings

In this section, we intend to provide comprehensive and concrete details about the task settings adopted in the experiment section of our paper. Specifically, we mainly introduce the dataset (i.e. Meta-Dataset [49]), and the vary-way vary-shot classification task setting in this section.

### D.1    Introduction of Meta-Dataset

Meta-Dataset is a dataset composed of several datasets that are easy and free to access. It spans a variety of visual concepts with different degrees of fine grain (e.g. from hand-writing to natural scenarios). It was first proposed by Triantafillou et al. [49] for cross-domain few-shot classification tasks and has been a popular and mainstream benchmark in this field. Originally, Meta-Dataset only contained 10 datasets, which are ILSVRC_2012 (a.k.a. ImageNet) [44], Omniglot [26], FGVC_Aircraft (Aircraft) [36], CUB_200-2011 (CU_Birds) [53], Describable Textures (DTD/Textures) [9], Qucik Draw [22], FGVCx Fungi (Fungi) [45], VGG_Flower [39], Traffic Sign [20], MSCOCO [33]. Later, MNIST [27], CIFAR-10 [25] and CIFAR-100 [25] are further contained by Bateni et al. [3].

In practice, there are two mainstream settings widely applied in the existing works [49, 43, 3, 10, 12, 34, 29, 30]. The first setting is the "train on all datasets" setting. In this setting, 8 datasets, including ImageNet, Omniglot, Aircraft, Birds, Textures, Quick Draw, Fungi, and VGG_Flower, are selected as seen domains while the remaining datasets are treated as unseen domains. In contrast, the other setting, the "train on ImageNet only" setting, only takes ImageNet as the seen domain while all other datasets are treated as unseen domains. In this paper, we report results under both task settings. In particular, if there is no further declaration, the default task setting is the "train on all datasets" setting.

### D.2    Brief Introduction of Vary-Way Vary-Shot Task Settings

Vary-way vary-shot classification task setting is first proposed in Meta-Dataset [49] to mimic the most common situations, where the number of classes and the number of samples in each class are randomly determined, in the real world. Thus, the samples in these tasks are imbalancedly distributed. Such task setting is widely adopted in many previous works [43, 3, 10, 50, 12, 34, 29, 30, 16]. Different from the conventional few-shot classification task setting, where the numbers of classes and shots are invariant in each task, vary-way vary-shot task setting adopted in cross-domain few-shot classification is much more challenging due to the imbalanced data and the uncertain task structure.

Roughly, the generation process of vary-way vary-shot task mainly includes two steps: class sampling and data point sampling. To be specific, the first step is sampling classes from a specific dataset. Given a dataset, the number of classes is randomly determined with some probability distribution in the interval $[5, N_{\max}]$, where $N_{\max}$ denotes the maximum number of classes of the dataset. In the context of Meta-Dataset, the $N_{\max}$ is usually set either 50 or as many classes as there are available.

After the number of classes is determined, a set of classes is uniformly sampled from the given dataset. Then, the number of data points for each class is assigned in a random way. Concretely, the

Table 4: Comparisons of CoPA respectively with linear transformation head and visual Transformation under both "train on all datasets" and "train on ImageNet only" settings. Mean accuracy and 95% confidence interval are reported. All results are the average of 5 reproductions with seeds 41-45.

| Datasets | train on all datasets | | train on ImageNet only | |
| --- | --- | --- | --- | --- |
| | CoPA + Linear | CoPA + ViT | CoPA + Linear | CoPA + ViT |
| ImageNet | 57.8±1.1 | 57.7±1.1 | 57.7±1.1 | 57.4±1.1 |
| Omniglot | 94.3±0.5 | 94.3±0.4 | **70.9±1.2** | 70.5±1.2 |
| Aircraft | 88.8±0.5 | 88.8±0.5 | **61.6±1.0** | 59.7±1.0 |
| Birds | 80.8±0.8 | 80.8±0.8 | 74.2±0.9 | 74.0±1.0 |
| Textures | **77.8±0.7** | 77.4±0.7 | **77.0±0.7** | 76.6±0.7 |
| Quick Draw | 82.8±0.6 | 82.8±0.6 | **61.3±1.0** | 58.7±1.0 |
| Fungi | 69.5±1.0 | 69.4±1.0 | **48.0±1.1** | 47.2±1.1 |
| VGG Flower | 92.7±0.5 | 92.6±0.5 | **88.9±0.6** | 88.3±0.7 |
| Traffic Sign | 66.6±1.1 | **68.8±1.1** | 63.8±1.1 | **68.2±1.1** |
| MSCOCO | **56.3±1.1** | 55.8±1.1 | **56.1±1.0** | 55.3±1.1 |
| MNIST | 95.2±0.4 | 95.2±0.5 | 87.3±0.7 | **89.7±0.7** |
| CIFAR-10 | **73.0±0.8** | 72.7±0.8 | 72.4±0.8 | 72.1±0.8 |
| CIFAR-100 | 63.4±1.0 | 63.2±1.0 | **62.7±1.0** | 62.3±1.0 |

---

**Algorithm 1 CoPA Algorithm**.

---

**Input:** pre-trained backbone $f_{\phi^*}$, number of inner iterations $n$, learning rate $\eta$, linear transformation heads $h_{\theta_P}$ and $h_{\theta_I}$, temperature coefficient $\tau$.
**Output:** the optimal parameters for linear transformation heads $\theta_P^*$ and $\theta_I^*$.
*# Sample a task*
**Sample** a new support data set $\mathcal{D}_{\mathcal{T}} = \{X, Y\}$;
**Generate** pseudo labels $Y_{\text{pseudo}} = \{0, 1, ..., |\mathcal{D}_{\mathcal{T}}| - 1\}$;
*# Performing contrastive prototype-image adaptation*
**for** $i = 1$ **to** $n$ **do**
    **Obtain** the prototype and instance representations:
        $Z_P = h_{\theta_P}(YY^\top f_{\phi^*}(X))$;
        $Z_I = h_{\theta_I}(f_{\phi^*}(X))$;
    **Compute** SCE loss $\mathcal{L}(\theta_P, \theta_I)$ in Eq. (3);
    **Update** parameters:
        $\theta_P \leftarrow \theta_P - \eta \nabla_{\theta_P} \mathcal{L}(\theta_P, \theta_I)$;
        $\theta_I \leftarrow \theta_I - \eta \nabla_{\theta_I} \mathcal{L}(\theta_P, \theta_I)$;
**end for**

---

first step in data point sampling is determining the number of query data. In the vary-way vary-shot task setting proposed in Meta-Dataset [49], the number of query data for each class is fixed to the same number (e.g. 10). The number of query data ought to be no more than half of all available data in the given class so that at least 50% of the data can be preserved as support data. Then, based on the number of query data, the support data are sampled from the remaining data with the constraint that the total number of the entire support data (of all classes) should be no more than 500. The number of support data for each class is determined by the size of support data and a set of random scalars sampled from a given interval. In this way, a task with random numbers of classes and shots is sampled from a given dataset. More concrete details are available in the Meta-Dataset article [49].

## E   Detailed Implementation Settings

In this section, we provide detailed implementation settings, including backbone pre-training, model structures, adaptation module initialization, hardware information, and other hyperparameter settings (e.g. learning rate & weight decay), for the convenience of reproducing our proposed CoPA method.

### E.1   Pre-trained Backbones

In this paper, our proposed CoPA method is built upon a ResNet-18 backbone [18] as done in previous works [12, 34, 29]. For the pre-trained backbone used in the "train on ImageNet only" setting, the

backbone is pre-trained in the same way as proposed in SUR [12]. Specifically, the backbone is trained with stochastic gradient descent optimizer and cosine annealing with a learning rate $3 \times 10^{-2}$. The momentum is set to 0.9 and the weight decay is set to $7 \times 10^{-4}$. In addition, the batch size is 64 and the total number of learning iterations is 480,000. All images are reshaped to the size of 84×84.

Moreover, for the universal backbone that is used in the "train on all datasets" setting, we follow URL [29] to distill a ResNet-18 backbone from models that are respectively pre-trained on 8 single datasets. The distilled model is also pre-trained with SGD optimizer (learning rate is set to $3 \times 10^{-2}$ and weight decay is set to $7 \times 10^{-4}$) and cosine annealing as well with 240,000 learning iterations. The details of single domain backbone pre-training are available in appendices of URL [29].

### E.2 Architecture of Visual Transformer

In addition to the linear transformation head, we also evaluate our method on visual Transformer (i.e. ViT) [11] to figure out whether the proposed CoPA method benefits from complex and powerful learning modules. In our experimental setting, the visual Transformer we adopted is composed of a single self-attention block. Specifically, each visual Transformer model only contains one attention block followed by a LayerNorm layer [1] and a linear layer. All layers in the ViT, including query, key, value heads, and the linear transformation head, are formulated as $512 \times 512$ matrices.

### E.3 Module Initialization

During the meta-test phase, the adaptation modules (i.e. trainable learning modules) are re-initialized for the adaptation of new tasks at the beginning of each episode. Specifically, all linear layers will be re-initialized to an identity matrix with the size of $512 \times 512$. Besides, in the visual Transformer case, the weight and bias of the LayerNorm layer are respectively set to 1.0 and 0.0, all class tokens are re-initialized to 0.0, and all position embeddings are re-initialized with a normal distribution $\mathcal{N}(0, 0.02)$. In particular, in CoPA + TSA case, the trainable learning modules plugged into the pre-trained backbone are re-initialized to all-ones matrices with a scale coefficient $1e - 4$.

### E.4 Optimization

In our method, the optimizer adopted during adaptation is Adam optimizer [24]. For each adaptation episode, the total number of iterations is 50 for linear transformation and 40 for visual Transformer. For linear head, the learning rate is set to 5e-3 for Traffic Sign and MNIST datasets while 1e-3 for the remaining datasets. The weight decay is set to 0.1 for all datasets except for Traffic Sign and MNIST. For the visual Transformer, the learning rate is set to 1e-3 for Traffic Sign and MNIST and 1e-4 for other datatsets. The weight decay is set to 0.1 for all datasets except Traffic Sign and MNIST.

### E.5 Hardware & Random Seeds

All experiments in this paper are conducted with an NVIDIA GeForce RTX 3090 GPU. To make sure that all experimental results are reproducible and the comparisons are fair, all results of our proposed CoPA method and other reproduced methods are the average of 5 reproductions with the random seeds 41, 42, 43, 44, 45. The GPU memory that is required for running our method is about 10 GB.

### E.6 Task Specific Adapter Setting

Among all existing methods, as far as we know, the best results on cross-domain few-shot classification tasks with respect to Meta-Dataset are reported by TSA [30]. Besides, recent work, TA$^2$-Net [16], can be seen as a further extension of TSA by adaptively selecting learning modules plugged into the pre-trained backbone. Due to the extra trainable modules plugged into the frozen pre-trained backbone, more task-specific knowledge can be learned to achieve better generalization performance.

In this paper, although our proposed CoPA method is able to achieve better generalization performance than TSA on seen domains without having to add any extra learning modules on the pre-trained backbone, there still exist obvious performance gaps on unseen domains. We conjecture that the improvements over unseen domains significantly rely on the extra learning modules. Thus, to verify this conjecture, we followed TSA to plug some extra learning modules into the pre-trained backbone to combine our proposed CoPA with TSA strategy. However, since TSA consumes large amounts

of time compared with URL in practice (about 30s/iter), concerning efficiency, we only plugged learning modules on the second and third residual blocks of the pre-trained ResNet-18 backbone.

In our implementation of CoPA + TSA case, we optimize the learning modules and transformation heads with Adam optimizer as well. To be specific, the learning rate of the two transformation heads is set to $5e - 3$ for Traffic Sign, MNIST, CIFAR-10, and CIFAR-100 datasets while $1e - 3$ for the remaining datasets. For learning modules plugged into the pre-trained backbones, the learning rate is set to $2.5e - 3$ for Traffic Sign, MNIST, CIFAR-10 and CIFAR-100 datasets while $1e - 4$ for the remaining datasets. Except for Traffic Sign and MNIST, the weight decay is set to 0.1.

### E.7 Reproduction of *State-of-the-art* Methods

In this paper, for fair comparisons between our proposed CoPA method and current *state-of-the-art* methods, such as URL, TSA, and TA$^2$-Net, we ran both CoPA and those existing methods with the same 5 random seeds and reported the average of 5 reproduction experiments as the final results. All results of these methods reported in our paper can be reproduced by setting the seed in the bash file.

## F More Experimental Results

### F.1 CoPA with More Complex Modules

Similar to CLIP [42], which performs contrastive learning via two Transformers [51, 11] with text and image data pairs, the proposed CoPA method in this paper fine-tunes two different modules respectively for prototype and image instance embeddings by substituting the text prompts with prototypes. However, the modules adopted in the main results are two linear transformations. It is not clear whether applying more complex modules contributes to the performance. Thus, in this section, we similarly evaluate the performance of CoPA with two visual Transformer (ViT) heads [11].

The implementation details of ViT are available in Appendix E.2. The comparison of performance between CoPA + ViT and CoPA + Linear is reported in Table 4. According to the results in the table, we can observe that the performance changes between CoPA + Linear and CoPA + ViT are slight under both "train on all datasets" and "train on ImageNet only" settings. Such a phenomenon reveals that the proposed CoPA method is not sensitive to the complexity of model architectures.

### F.2 Fewer Shot Classification Tasks.

In this section, we follow previous works [49] to evaluate our proposed CoPA method on fewer-shot classification tasks. Different from the vary-way vary-shot task setting mentioned above, the fewer-shot classification task contains fewer data samples in each task and thus is more challenging.

#### F.2.1 Fewer-shot Experiments

In addition to the vary-way vary-shot task setting adopted in the main results section, some other task settings, such as 5-way 1-shot and vary-way 5-shot, are also widely adopted in previous works [14, 47, 49, 29, 30]. Compared with the vary-way vary-shot task setting, 5-way 1-shot and vary-way 5-shot tasks are more challenging since fewer data samples are included in the support set of each task. For example, in the vary-way 5-shot task setting, although the number of classes is still randomly determined at the beginning of each episode, the number of data samples (i.e. shot) is fixed to 5. Similarly, in the 5-way 1-shot task setting, both the numbers of classes and data samples in each class are respectively fixed to 5 and 1. In this section, in order to further explore the capability of CoPA, we validate the performance of CoPA on both task settings aforementioned under the "train on all datasets" setting for all datasets in Meta-Dataset. All results are reported in Tables 5 and 6.

**Vary-way 5-shot.** As shown in Table 5, compared with previous works, such as SimpleCNAPs [3], SUR [12], URT [29] and URL [29], CoPA achieves the best performance on 7 out of 13 datasets and ranks 3.2 among all approaches. In particular, CoPA outperforms URL, which CoPA is based on, on 9 out of 13 datasets, and achieves 0.2%, 0.5%, and 0.3% improvements on average respectively on seen, unseen, and all domain cases. To be specific, CoPA respectively achieves better performance on Omniglot (0.2%), Aircraft (0.4%), CU_Birds (0.2%), VGG_Flower (0.5%) and Traffic Sign (2.1%) datasets. Meanwhile, we also notice that CoPA achieves better results than TSA, where extra trainable

Table 5: **Results under vary-way 5-shot task setting (under "Trained on all datasets" setting).** Mean accuracy, 95% confidence interval reported.

| Datasets | Vary-way 5-shot | | | | | | |
|---|---|---|---|---|---|---|---|
| | Sim-CNAPS | SUR | URT | URL | CoPA | TSA | CoPA + TSA |
| **ImageNet** | 47.2±1.0 | 46.7±1.0 | **48.6±1.0** | 47.8±1.0 | 47.6±1.0 | 47.0 ± 1.0 | 47.1 ± 1.0 |
| **Omniglot** | 95.1±0.3 | 95.8±0.3 | **96.0±0.3** | 95.8±0.3 | **96.0±0.3** | 96.4 ± 0.3 | 96.4 ± 0.3 |
| **Aircraft** | 74.6±0.6 | 82.1±0.6 | 81.2±0.6 | 83.9±0.5 | 84.3±0.5 | 84.6 ± 0.5 | 84.6 ± 0.5 |
| **Birds** | 69.6±0.7 | 62.8±0.9 | 71.2±0.7 | 76.1±0.7 | **76.3±0.6** | 76.3 ± 0.7 | 76.6 ± 0.6 |
| **Textures** | 57.5±0.7 | 60.2±0.7 | 65.2±0.7 | 66.8±0.6 | **66.9±0.6** | 66.4 ± 0.6 | 66.7 ± 0.6 |
| **Quick Draw** | 70.9±0.6 | 79.0±0.5 | **79.2±0.5** | 78.3±0.5 | 78.6±0.5 | 78.1 ± 0.5 | 78.3 ± 0.5 |
| **Fungi** | 50.3±1.0 | 66.5±0.8 | 66.9±0.9 | 68.7±0.9 | **68.8±0.9** | 68.5 ± 0.9 | 68.5 ± 0.9 |
| **VGG Flower** | 86.5±0.4 | 76.9±0.6 | 82.4±0.5 | 88.5±0.4 | **89.0±0.4** | 89.3 ± 0.4 | 89.3 ± 0.4 |
| **Traffic Sign** | 55.2±0.8 | 44.9±0.9 | 45.1±0.9 | 56.7±0.8 | **58.8±0.8** | 71.5 ± 0.5 | 80.0 ± 0.5 |
| **MSCOCO** | 49.2±0.8 | 48.1±0.9 | 52.3±0.9 | 51.3±0.8 | 50.9±0.8 | 52.0 ± 0.9 | 52.4 ± 0.8 |
| **MNIST** | 88.9±0.4 | 90.1±0.4 | 86.5±0.5 | 88.5±0.4 | 89.5±0.4 | 92.2 ± 0.3 | 94.8 ± 0.3 |
| **CIFAR-10** | 66.1±0.7 | 50.3±1.0 | 61.4±0.7 | 59.6±0.7 | 59.6±0.7 | 66.3 ± 0.7 | 65.7 ± 0.7 |
| **CIFAR-100** | 53.8±0.9 | 46.4±0.9 | 52.5±0.9 | 55.8±0.9 | 55.3±0.8 | 62.2 ± 0.8 | 63.1 ± 0.7 |
| **Average Seen** | 69.0 | 71.2 | 73.8 | 75.7 | **75.9** | 75.8 | **75.9** |
| **Average Unseen** | 62.6 | 56.0 | 59.6 | 62.3 | **62.8** | 68.8 | **71.2** |
| **Average All** | 66.5 | 65.4 | 68.3 | 70.6 | **70.9** | 73.1 | **74.1** |
| **Rank** | 5.6 | 5.9 | 4.4 | 3.9 | **3.2** | 2.8 | **2.2** |

[1] Both the results on URL, TSA and CoPA are the average of 5 random seeds (41 - 45).

Table 6: **Results under 5-way 1-shot task settings (under "Trained on all datasets" setting).** Mean accuracy, 95% confidence interval reported.

| Datasets | 5-way 1-shot | | | | | | |
|---|---|---|---|---|---|---|---|
| | Sim-CNAPS | SUR | URT | URL | CoPA | TSA | CoPA + TSA |
| **ImageNet** | 42.6±0.9 | 40.7±1.0 | **47.4±1.0** | 46.5±1.0 | 46.5±1.1 | 46.1 ± 1.0 | 46.3 ± 1.0 |
| **Omniglot** | 93.1±0.5 | 93.0±0.7 | 95.6±0.5 | 95.5±0.5 | 95.5±0.5 | 95.6 ± 0.5 | **95.7 ± 0.5** |
| **Aircraft** | 65.8±0.9 | 67.1±1.4 | 77.9±0.9 | 78.6±0.9 | 78.4±0.9 | 78.6 ± 0.9 | **78.9 ± 0.9** |
| **Birds** | 67.9±0.9 | 59.2±1.0 | 70.9±0.9 | **76.2±0.9** | **76.2±0.9** | 75.8 ± 0.9 | 75.8 ± 0.9 |
| **Textures** | 42.2±0.8 | 42.5±0.8 | 49.4±0.9 | 52.0±0.9 | **52.2±0.9** | 51.9 ± 0.9 | 52.0 ± 0.9 |
| **Quick Draw** | 70.5±0.9 | **79.8±0.9** | 79.6±0.9 | 79.1±0.9 | 79.1±0.9 | 78.9 ± 0.9 | 79.0 ± 0.9 |
| **Fungi** | 58.3±0.1 | 64.8±1.1 | 71.0±1.0 | **71.4±1.0** | **71.4±1.0** | 71.3 ± 1.0 | 71.4 ± 1.0 |
| **VGG Flower** | 79.9±0.7 | 65.0±1.0 | 72.7±1.0 | 80.3±0.8 | 80.3±0.8 | 80.5 ± 0.8 | **80.6 ± 0.8** |
| **Traffic Sign** | 55.3±0.9 | 44.6±0.9 | 52.7±0.9 | 57.4±0.9 | 58.4±0.9 | 57.3 ± 0.9 | 57.8 ± 1.0 |
| **MSCOCO** | 48.8±0.9 | 47.8±1.1 | **56.9±1.1** | 52.1±1.0 | 51.7±0.9 | 52.9 ± 1.0 | 52.1 ± 1.0 |
| **MNIST** | **80.1±0.9** | 77.1±0.9 | 75.6±0.9 | 73.3±0.8 | 73.4±0.8 | 75.1 ± 0.8 | 76.1 ± 0.8 |
| **CIFAR-10** | 50.3±0.9 | 35.8±0.8 | 47.3±0.9 | 48.6±0.8 | 48.4±0.8 | 49.2 ± 0.8 | **54.5 ± 0.8** |
| **CIFAR-100** | 53.8±0.9 | 42.9±1.0 | 54.9±1.1 | 61.5±1.0 | 61.3±1.0 | 62.4 ± 1.0 | **62.6 ± 1.0** |
| **Average Seen** | 65.0 | 64.0 | 70.6 | **72.5** | **72.5** | 72.3 | **72.5** |
| **Average Unseen** | 57.7 | 49.6 | 57.5 | 58.4 | **58.6** | 59.4 | **60.6** |
| **Average All** | 62.2 | 58.5 | 65.5 | **67.1** | **67.1** | 67.3 | **67.9** |
| **Rank** | 5.5 | 5.9 | 4.1 | **3.2** | **3.2** | 3.5 | **2.3** |

[1] Both the results on URL, TSA and CoPA are the average of 5 random seeds (41 - 45).

learning modules are plugged into the frozen pre-trained backbones. Concretely, CoPA outperforms TSA respectively on ImageNet (0.6%), Textures (0.5%), Quick Draw (0.4%), and Fungi (0.3%).

Moreover, we also evaluate the performance of CoPA + TSA on vary-way 5-shot tasks. According to the results in the table, CoPA + TSA achieves the best performance on 8 out of 13 datasets and ranks 2.2 among all approaches. In general, CoPA + TSA obtains the similar performance on seen domains to CoPA and TSA while significantly outperforms all other methods on unseen domains. Specifically, compared with the state-of-the-art method TSA, CoPA + TSA achieves 8.5%, 0.4%, 2.6%, and 0.9% improvements respectively on Traffic Sign, MSCOCO, MNIST, and CIFAR-100 datasets.

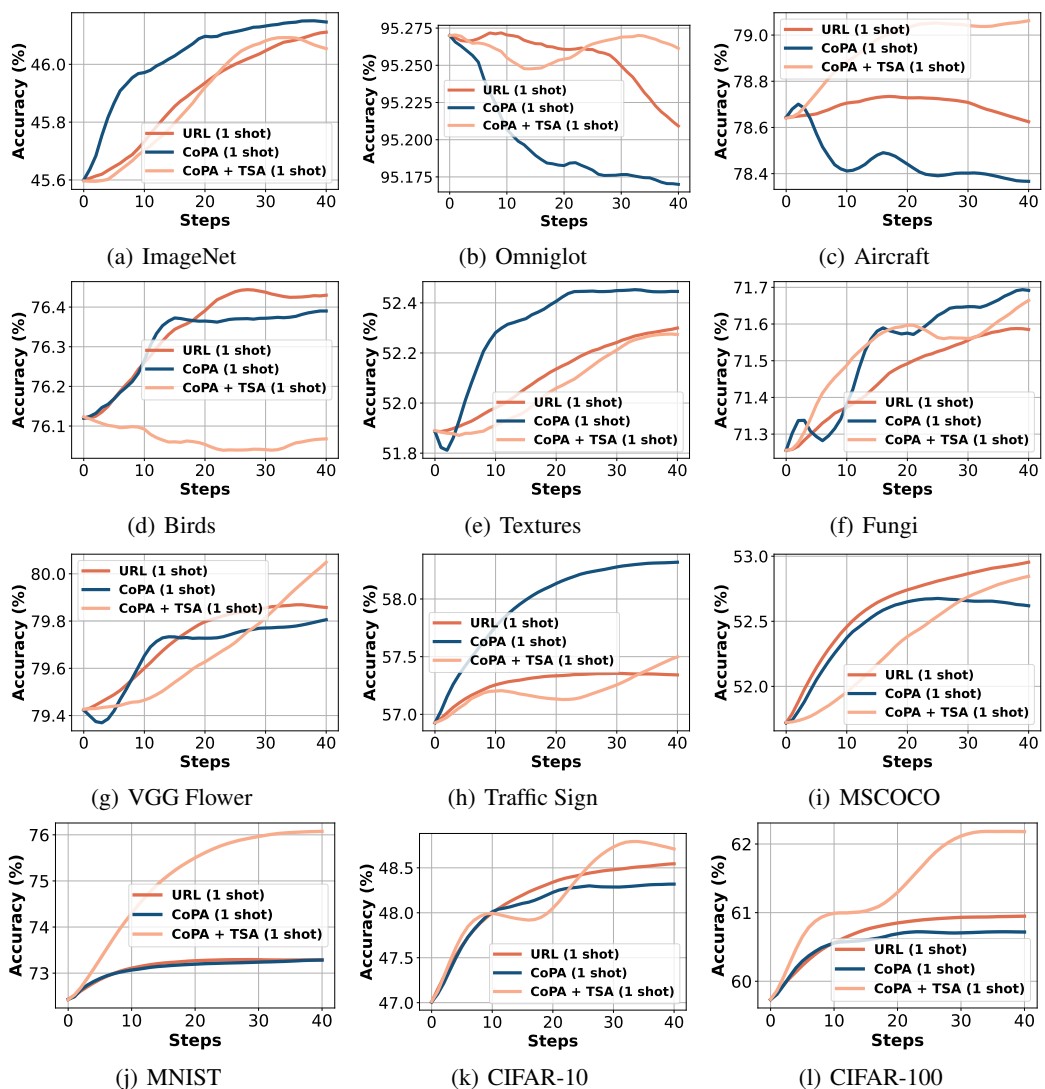

Figure 9: **Comparisons of test accuracy curves between URL, CoPA and CoPA + TSA under 5-way 1-shot task setting.** The figures depict the comparison of validation accuracy between URL, CoPA and CoPA + TSA under the 5-way 1-shot setting during the adaptation process. According to these figures, it is easy to observe that overfitting tends to take place in all cases in URL and several cases in CoPA during the adaptation process. The main reason is that the prototype of each class under the 5-way 1-shot setting is calculated with only one data sample. Thus, such prototypes are too biased to contain the comprehensive and useful information regarding the classes they belong to.

All these results demonstrate that our proposed CoPA is robust and able to achieve better generalization performance compared with the existing approaches on CFC tasks even with fewer data samples.

**5-way 1-shot.** As to 5-way 1-shot task, it is a special case in the few-shot classification task since the prototype of each class in this task setting is calculated with only one data sample. Thus, the obtained prototypes may be more biased compared with those obtained in the vary-way vary-shot and vary-way 5-shot task settings. Consequently, the 5-way 1-shot task is much more challenging.

According to the results reported in Table 6, it is easy to observe that our proposed CoPA achieves comparable results compared with URL and ranks 3.2 among all approaches. To be concrete, CoPA achieves the similar performance to URL on the seen domains while only outperforms URL on Traffic Sign dataset among the unseen domains. As for TSA, we find that plugging more extra trainable learning modules into the pre-trained backbone fails to improve the generalization performance on the seen domains. However, the performance on the unseen domains benefits from these extra learning modules. Specifically, although TSA fails to outperform our proposed CoPA on Traffic Sign

Table 7: **Study of the size of support set (Under "train on all dataset" settings).**

| Methods | ImageNet | Omniglot | Aircraft | Birds | DTD | QuickDraw | Fungi | VGG_Flower |
|---|---|---|---|---|---|---|---|---|
| Max | 498 | 165 | 497 | 494 | 498 | 497 | 494 | 497 |
| Min | 6 | 5 | 5 | 6 | 6 | 16 | 7 | 5 |
| Avg | 364.9 | 49.7 | 340 | 312.3 | 282.3 | 428.0 | 252.5 | 276.3 |

dataset, it achieves better performance on other unseen domain datasets compared with URL and CoPA. Moreover, in order to further explore our proposed CoPA method on 5-way 1-shot tasks, we further evaluate the combination of CoPA and TSA (i.e. "CoPA + TSA" in the table). According to the results reported in the table, we find that CoPA + TSA achieves the best performance on average among all approaches and ranks 2.3. Compared with the original TSA, CoPA + TSA improves the generalization performance on the seen domains and obtains comparable results to URL and CoPA. On unseen domains, CoPA + TSA performs better than TSA. For example, CoPA + TSA increases the performance on Traffic Sign, MNIST and CIFAR-10 respectively by $0.5\%$, $1.0\%$, and $5.3\%$.

One conjecture for the phenomenon that CoPA fails to outperform other approaches evidently in most cases is that the prototype of each class generated under the 5-way 1-shot setting fails to contain comprehensive and useful information that is representative enough to depict the general features of the class since it is calculated with only one data sample. In this case, contrastive learning is performed between two identical data samples and overfitting may take place. In order to validate our conjecture, we plot the accuracy curves of URL, CoPA, and CoPA + TSA. The visualization results are presented in Fig. 9. According to these figures, we can see that it is quite hard to perform adaptation on tasks with extremely few labeled data samples and overfitting tends to take place. For example, For datasets like Omniglot (Fig. 9(b)) and Aircraft (Fig. 9(c)), overfitting is obvious on both URL and CoPA. Meanwhile, the application of CoPA even deteriorates the performance.

### F.2.2 Study on Effect of Support Data Size

In this paper, due to the consideration of preserving the gap between prototypes and image instances, we propose to adopt symmetric cross-entropy loss as the learning objective. Typically, the symmetric cross-entropy loss is commonly applied in the contrastive learning framework.

An important issue in the conventional contrastive learning framework is the size of the data. As demonstrated in previous works [5], the large batch size is the key to learning useful representations for downstream tasks. Thus, we are also interested in the effect of the batch size of the data. However, due to the task settings, such as the vary-way vary-shot setting, it is intractable to manipulate the batch size via task sampling directly. Thus, we can approximately study the problem by comparing the performance of CoPA respectively on vary-way vary-shot tasks and fewer-shot tasks since the latter contains much fewer data samples in the support set.

First of all, we count the number of samples in 600 tasks randomly sampled from the validation set of seen domain datasets (with the random seed 42). The results are reported in Table 7. According to the table, we can observe that the difference between the minimum and the maximum of the support set size is large. On average, in most cases, the average size of the support set is more than 200. These results indicate that the number of data samples under vary-way vary-shot settings is not too small.

Then, we compared the performance of CoPA respectively on vary-way vary-shot tasks, vary-way 5-shot tasks, and 5-way 1-shot tasks. In vary-way vary-shot setting, which contains the most samples, CoPA evidently achieves the best results. However, in vary-way 5-shot tasks, CoPA can still achieve the best results on average, though it fails to outperform on some datasets. On 5-way 1-shot tasks, we can obviously observe that CoPA achieves similar results to URL without absolute advantages.

Thus, we can see that the rule is not changed and the large batch size is still preferred in our proposed method. Thus, a potential limitation of our proposed method is that CoPA may fail to achieve good generalization performance with extremely few data samples.

### F.3 Results of Analysis regarding Number of Parameters

In this paper, the original CoPA aims at fast fine-tuning two linear heads respectively for prototype and image instance embeddings on top of the frozen pre-trained backbones in the same way as contrastive

learning applied in CLIP [42]. Compared with previous works, such as URL [29] on which CoPA is based, CoPA utilizes more model parameters. To be specific, the number of parameters contained in the trainable modules of CoPA is twice those contained in the trainable module in the URL. Thus, in this case, a concern that may be raised is whether the improvements in generalization performance on CoPA result from more trainable parameters contained in the two linear transformation heads.

In order to figure out this question, we conduct an analysis to explore the effect of the number of model parameters. To be specific, we compare CoPA with a variant of URL that includes the same number of trainable model parameters under the "train on all datasets" setting. In the URL, only a simple linear layer with the size of $512 \times 512$ is trained on support data for further few-shot classification on query data. In order to increase the number of the number of parameters, we propose to substitute the original linear transformation head in the URL with a simple two-layer MLP. The MLP is designed with the structure of `Linear-Batch Normalization-Linear`. All linear layers in the MLP model are with the size of $512 \times 512$. Then, the number of parameters is equal to CoPA.

The experimental results are reported in Table 9. According to these results, we can observe that the combination of URL and MLP fails to outperform CoPA and even achieve worse results than URL where fewer parameters are contained. These results demonstrate that simply adding more parameters in the linear transformation head does not positively affect the generalization performance since the implicit assumption about the shared transformation is not removed. The success of CoPA is mainly based on the mechanism of CoPA, where the prototype and image instance representations are explored respectively with different transformations by optimizing symmetric cross entropy loss.

## F.4 Efficiency of CoPA.

Due to the two linear transformation heads, the number of parameters in CoPA is twice of that in URL. Assume that there are $N$ data samples with the dimension of $N_d$, the number of classes is $N_C$, then the size of a linear head is $N_d \times N_d$. For instance, in our experimental settings, $N_d = 512$ and the size of the linear head is $512 \times 512$. Based on this, we list the number of parameters and the computational complexity of a forward pass (including representation transformations with linear heads and inner products between samples and prototypes) in Table 8. Notice that we did not take the computation of feature extraction from the frozen backbone into consideration. However, as shown in Fig. 10, CoPA is more efficient than URL. The main reason for this phenomenon is that CoPA calculates the prototypes at the beginning of each episode and does not have to calculate prototypes repeatedly in each iteration as done in URL.

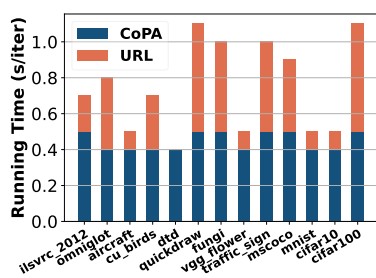

Figure 10: **The comparison of running time between URL and CoPA.**

Table 8: Comparison of the number of parameters, the computational complexity of URL and CoPA.

| Methods | # Params | Computational Complexity |
|---------|----------|--------------------------|
| URL | $N_d^2$ | $N(N_C + N_d)(N_d^2 - N_d)$ |
| CoPA | $2N_d^2$ | $(NN_d + N_C N_d + 2N^2)(N_d^2 - N_d)$ |

## F.5 Ablation Study

In this section, we perform a series of ablation studies to evaluate the abilities of our proposed CoPA method and determine the functions of the components, such as the SCE loss, adopted in the CoPA.

**Symmetry Cross Entropy Loss.** In our paper, we follow CLIP [42] and propose a simple yet effective method, contrastive prototype-image adaptation (CoPA), to perform model adaptation on prototype and image representation pairs via contrastive learning. The loss is the symmetry cross entropy (SCE) loss, which is widely adopted in contrastive learning [40, 56]. To validate the effect of SCE loss, we conducted an ablation study on SCE loss respectively on URL and CoPA methods.

To be specific, for URL, we replace the NCC loss with SCE loss and fine-tune the shared transformation head in the same way as URL. For CoPA, we substitute the SCE loss with NCC loss used in

Table 9: **Effect of the number of parameters (Under "Train on all dataset" settings).**

| Datasets | URL | URL + MLP | CoPA |
|---|---|---|---|
| **# Params** | $2^{18}$ | $2^{19}$ | $2^{19}$ |
| **ImageNet** | 57.3±1.1 | 56.1±1.1 | **57.8±1.1** |
| **Omniglot** | 94.1±0.4 | **94.6±0.4** | 94.3±0.5 |
| **Aircraft** | 88.2±0.5 | 86.6±0.5 | **88.8±0.8** |
| **Birds** | 80.2±0.7 | 79.4±0.7 | **80.8±0.8** |
| **Textures** | 76.2±0.7 | 72.6±0.7 | **77.8±0.7** |
| **Quick Draw** | 82.2±0.6 | 81.7±0.6 | **82.8±0.6** |
| **Fungi** | 68.7±1.0 | 66.7±1.0 | **69.5±1.0** |
| **VGG Flower** | 91.9±0.5 | 91.5±0.5 | **92.7±0.5** |
| **Traffic Sign** | 63.3±1.2 | 54.5±1.1 | **66.6±1.1** |
| **MSCOCO** | 54.2±1.0 | 53.1±1.0 | **56.3±1.1** |
| **MNIST** | 94.7±0.4 | 92.5±0.4 | **95.2±0.4** |
| **CIFAR-10** | 71.9±0.8 | 70.9±0.8 | **73.0±0.8** |
| **CIFAR-100** | 62.9±1.0 | 60.6±1.0 | **63.4±1.1** |

Table 10: Ablation study on symmetry cross entropy loss and different transformation heads (Under the "train on all dataset" setting).

| Datasets | URL | URL + SCE | URL + 2Heads | CoPA | CoPA + NCC |
|---|---|---|---|---|---|
| **ImageNet** | 57.3±1.1 | 57.2±1.1 | 56.5±1.1 | **57.8±1.1** | 52.5±1.0 |
| **Omniglot** | 94.1±0.4 | 94.1±0.4 | 94.1±0.5 | **94.3±0.5** | 93.8±0.5 |
| **Aircraft** | 88.2±0.5 | 87.6±0.5 | 88.3±0.5 | **88.8±0.8** | 85.6±0.5 |
| **Birds** | 80.2±0.7 | 80.1±0.7 | 80.0±0.8 | **80.8±0.8** | 78.2±0.7 |
| **Textures** | 76.2±0.7 | 75.9±0.7 | 76.8±0.7 | **77.8±0.7** | 71.1±0.7 |
| **Quick Draw** | 82.2±0.6 | 82.2±0.6 | 82.1±0.6 | **82.8±0.6** | 80.9±0.6 |
| **Fungi** | 68.7±1.0 | **69.4±1.0** | 64.0±1.0 | 69.5±1.0 | 57.9±0.8 |
| **VGG Flower** | 91.9±0.5 | 91.9±0.5 | 91.3±0.5 | **92.7±0.5** | 85.0±0.6 |
| **Traffic Sign** | 63.3±1.2 | 62.3±1.2 | 62.0±1.1 | **66.6±1.1** | 41.5±1.2 |
| **MSCOCO** | 54.2±1.0 | **55.9±1.0** | 51.8±1.0 | 56.3±1.1 | 37.6±1.1 |
| **MNIST** | 94.7±0.4 | **94.9±0.4** | 94.7±0.5 | 95.2±0.4 | 93.4±0.5 |
| **CIFAR-10** | 71.9±0.8 | **72.5±0.8** | 71.7±0.8 | 73.0±0.8 | 65.0±0.7 |
| **CIFAR-100** | 62.9±1.0 | 62.8±1.0 | 62.7±1.0 | **63.4±1.1** | 58.4±0.9 |

previous frameworks [29, 30] and evaluate the invariant algorithm. Both experiments are conducted under the "train on all datasets" setting on Meta-Dataset. The hyperparameter settings in the variant CoPA algorithm are consistent with those in the CoPA. For fairness, all the results are the average of 5 experiments with different random seeds. The numerical results are reported in Table 10.

According to the table, on the one hand, URL+SCE loss achieves comparable performance on most datasets, such as ImageNet and Omniglot. An interesting phenomenon is URL+SCE achieves significantly better performance on some difficult datasets, such as Fungi and MSCOCO, where URL tends to overfit the data. This observation indicates that SCE loss can alleviate the constraints to some extent, but the assumption of the shared representation transformation still constrains its capability. We think such a phenomenon is reasonable. In typical contrastive learning, such as SimCLR [5] and BYOL [15], siamese networks are adopted to respectively perform representation encoding in case of the collapse of the networks. Thus, the shared transformation may be naturally not good at performing representation learning on two sets of data. Meanwhile, the performance of CoPA + NCC drops significantly compared to CoPA. Thus, both two observations demonstrate that SCE loss facilitates achieving better generalization performance in transformation model adaptation.

**Different Representation Transformations.** As aforementioned, previous works [29, 30] implicitly assume that the same representation transformation is shared between prototype and image instance embeddings. However, in this paper, both empirical results and analyses indicate that such an assumption constrains the transformation models from learning desirable representations. To solve the problem, we propose to simply apply different transformations respectively for prototypes and images. In this part, we perform an ablation study to validate the effect of the different transformations.

According to the reported results in the table, although better performance is achieved on some datasets, such as Textures, URL+2Heads achieves comparable or even worse performance in other cases. In order to further examine the reason for such a phenomenon, we plot the test loss curves in Fig. 6. According to the figures, we can observe that overfitting takes place during the adaptation phase. The overfitting phenomenon, to some extent, reflects that discriminative representations are learned via two different transformations. This phenomenon conforms to our intuition that the application of different transformations contributes to learning more discriminative representations.

Thus, with all these aspects taken into consideration, we find that the different transformations help learn more discriminative representations respectively for prototypes and image instances. Meanwhile, the SCE loss contributes to improving generalization performance. The combination of these two techniques collaboratively facilitates achieving impressive empirical results on Meta-Dataset.

# G    Proof Results

In this section, we provide detailed proof of the theoretical results in the main paper.

## G.1    Proof of Theorem 3.1

*Proof.* With Eq. (1), by replacing $d(\cdot, \cdot)$ with cosine similarity, we have:

$$
\begin{aligned}
\mathcal{L}(\theta) &= -\frac{1}{n} \sum_{i=1}^{n} \log \frac{e^{\boldsymbol{z}_i^\top \cdot \boldsymbol{c}_c}}{\sum_{j=1}^{N_C} e^{\boldsymbol{z}_i^\top \cdot \boldsymbol{c}_j}} \\
&= -\frac{1}{n} \sum_{i=1}^{n} \boldsymbol{z}_i^\top \boldsymbol{c}_c - \frac{1}{n} \sum_{i=1}^{n} \left[ -\log \sum_{j=1}^{N_C} e^{\frac{1}{|\mathcal{C}_j|} \sum_{\boldsymbol{z}' \in \mathcal{C}_j} \boldsymbol{z}_i^\top \boldsymbol{z}'} \right] \\
&= -\frac{1}{n} \sum_{i=1}^{n} \boldsymbol{z}_i^\top \boldsymbol{c}_c - \frac{1}{n} \sum_{i=1}^{n} \left[ -\log \frac{1}{N_C} \sum_{j=1}^{N_C} e^{\frac{1}{|\mathcal{C}_j|} \sum_{\boldsymbol{z}' \in \mathcal{C}_j} \boldsymbol{z}_i^\top \boldsymbol{z}'} - \log N_C \right] \\
&\geq -\frac{1}{n} \sum_{i=1}^{n} \boldsymbol{z}_i^\top \boldsymbol{c}_c + \frac{1}{n} \sum_{i=1}^{n} \sum_{j=1}^{N_C} \sum_{\boldsymbol{z}' \in \mathcal{C}_j} \frac{1}{N_C |\mathcal{C}_j|} \boldsymbol{z}_i^\top \boldsymbol{z}',
\end{aligned}
$$

where $\boldsymbol{z}'$ is the independent copy of $\boldsymbol{z}$, $\mathcal{C}_j$ denotes the set of sample representations of class $j$. The inequality sign is derived from Jesen's Inequality. Then, with a constant that satisfies $0 \leq \alpha < \frac{1}{N_C |\mathcal{C}_j|}$ for $\forall j$, we further have:

$$
\mathcal{L}(\theta) \geq -\frac{1}{n} \sum_{i=1}^{n} \boldsymbol{z}_i^\top \boldsymbol{c}_c + \frac{\alpha}{n} \sum_{i=1}^{n} \sum_{\boldsymbol{z}' \in \mathcal{Z}} \boldsymbol{z}_i^\top \boldsymbol{z}'.
$$

## G.2    Proof of the Bound of the Gaps

**Lemma G.1.** *Consider a square matrix $\Theta = [\Theta^1, \Theta^2, ..., \Theta^d] \in \mathbb{R}^{d \times d}$, where $\Theta^i \in \mathbb{R}^d$ is the $i$-th column of $\Theta$, and a vector $\boldsymbol{v} \in \mathbb{R}^d$. Then, we have:*

$$
\min_{1 \leq i \leq d} \cos^2(\boldsymbol{v}, \Theta^i) \|\boldsymbol{v}\|_2^2 \|\Theta\|_F^2 \leq \left\| \boldsymbol{v}^\top \Theta \right\|_2^2 \leq \max_{1 \leq j \leq d} \cos^2(\boldsymbol{v}, \Theta^j) \|\boldsymbol{v}\|_2^2 \|\Theta\|_F^2 ,
$$

*where $|| \cdot ||_F$ and $|| \cdot ||_2$ respectively denote the Frobenius and $L_2$ norm and $\cos(\cdot, \cdot)$ denotes the cosine similarity function.*

*Proof.* We first expand the $\boldsymbol{v}^\top \Theta$ term:

$$\left\|\boldsymbol{v}^\top \Theta\right\|_2^2 = \left\|\boldsymbol{v}^\top \left[\Theta^1, \Theta^2, ..., \Theta^d\right]\right\|_2^2$$
$$= \left\|\left[\boldsymbol{v}^\top \Theta^1, \boldsymbol{v}^\top \Theta^2, ..., \boldsymbol{v}^\top \Theta^d\right]\right\|_2^2$$
$$= \sum_{i=1}^d \left(\boldsymbol{v}^\top \Theta^i\right)^2$$
$$= \|\boldsymbol{v}\|_2^2 \sum_{i=1}^d \left\|\Theta^i\right\|_2^2 \cos^2(\boldsymbol{v}, \Theta^i).$$

Then, we can obtain the lower and upper bound of $\left\|\boldsymbol{v}^\top \Theta\right\|_2^2$:

$$\min_{1 \le i \le d} \cos^2(\boldsymbol{v}, \Theta^i) \|\boldsymbol{v}\|_2^2 \sum_{i=1}^d \left\|\Theta^i\right\|_2^2 \le \left\|\boldsymbol{v}^\top \Theta\right\|_2^2 \le \max_{1 \le j \le d} \cos^2(\boldsymbol{v}, \Theta^j) \|\boldsymbol{v}\|_2^2 \sum_{i=1}^d \left\|\Theta^j\right\|_2^2.$$

Since $\sum_{i=1}^d \left\|\Theta^i\right\|_2^2 = \sum_{i=1}^d \sum_{k=1}^d (\Theta^{ik})^2 = \|\Theta\|_F^2$, we further have:

$$\min_{1 \le i \le d} \cos^2(\boldsymbol{v}, \Theta^i) \|\boldsymbol{v}\|_2^2 \|\Theta\|_F^2 \le \left\|\boldsymbol{v}^\top \Theta\right\|_2^2 \le \max_{1 \le j \le d} \cos^2(\boldsymbol{v}, \Theta^j) \|\boldsymbol{v}\|_2^2 \|\Theta\|_F^2.$$

$\square$

### G.2.1 Proof of Theorem 3.2

*Proof.* First of all, we can observe that $\frac{1}{|\mathcal{D}_\mathcal{T}|} \sum_{\boldsymbol{z} \in \mathcal{Z}} \boldsymbol{z}$ and $\frac{1}{N_C} \sum_{\boldsymbol{c} \in \mathcal{C}} \boldsymbol{c}$ are $d$-dimension vectors. Then, we can expand the representation gap:

$$\left\|\frac{1}{|\mathcal{D}_\mathcal{T}|} \sum_{\boldsymbol{z} \in \mathcal{Z}} \boldsymbol{z} - \frac{1}{N_C} \sum_{\boldsymbol{c} \in \mathcal{C}} \boldsymbol{c}\right\|_2^2 = \left\|\left[\frac{1}{|\mathcal{D}_\mathcal{T}|} \sum_{\boldsymbol{x} \in \mathcal{D}_\mathcal{T}} f_{\phi^*}(\boldsymbol{x}) - \frac{1}{N_C} \sum_{b=1}^{N_C} \left(\frac{1}{|\mathcal{C}_b|} \sum_{\boldsymbol{x}' \in \mathcal{C}_b} f_{\phi^*}(\boldsymbol{x}')\right)\right]^\top \Theta\right\|_2^2.$$

Then, with Lemma G.1, we have the bounds of the gap between the prototype and image instance representations:

$$m \|\Theta\|_F^2 \left\|\vec{\Delta}_{\text{emb}}\right\|_2^2 \le \left\|\frac{1}{|\mathcal{D}_\mathcal{T}|} \sum_{\boldsymbol{x} \in \mathcal{D}_\mathcal{T}} \boldsymbol{z} - \frac{1}{N_C} \sum_{\boldsymbol{c} \in \mathcal{C}} \boldsymbol{c}\right\|_2^2 \le M \|\Theta\|_F^2 \left\|\vec{\Delta}_{\text{emb}}\right\|_2^2,$$

where $\vec{\Delta}_{\text{emb}} = \frac{1}{|\mathcal{D}_\mathcal{T}|} \sum_{i=1}^{|\mathcal{D}_\mathcal{T}|} f_{\phi^*}(\boldsymbol{x}_i) - \frac{1}{N_C} \sum_{b=1}^{N_C} \left(\frac{1}{|\mathcal{C}_b|} \sum_{\boldsymbol{x}' \in \mathcal{C}_b} f_{\phi^*}(\boldsymbol{x}')\right)$ denotes the gap between embeddings, $m = \min_{1 \le i \le d} \cos^2(\vec{\Delta}_{\text{emb}}, \Theta^i)$ and $M = \max_{1 \le j \le d} \cos^2(\vec{\Delta}_{\text{emb}}, \Theta^j)$. $\square$

### G.3 Proof of Theorem 5.1

*Proof.* We first calculate the first term of Eq. (4). Then, we have:

$$-\frac{1}{|\mathcal{D}_\mathcal{T}|} \sum_{i=1}^{|\mathcal{D}_\mathcal{T}|} \log \frac{\exp(\boldsymbol{z}_i^\top \boldsymbol{c}_i)}{\sum_{j=1}^{|\mathcal{D}_\mathcal{T}|} \exp(\boldsymbol{z}_i^\top \boldsymbol{c}_j)} = -\frac{1}{|\mathcal{D}_\mathcal{T}|} \sum_{i=1}^{|\mathcal{D}_\mathcal{T}|} \boldsymbol{z}_i^\top \boldsymbol{c}_i + \frac{1}{|\mathcal{D}_\mathcal{T}|} \sum_{i=1}^{|\mathcal{D}_\mathcal{T}|} \log \sum_{j=1}^{|\mathcal{D}_\mathcal{T}|} \exp(\boldsymbol{z}_i^\top \boldsymbol{c}_j)$$

$$= -\frac{1}{|\mathcal{D}_\mathcal{T}|} \sum_{i=1}^{|\mathcal{D}_\mathcal{T}|} \boldsymbol{z}_i^\top \boldsymbol{c}_i + \frac{1}{|\mathcal{D}_\mathcal{T}|} \sum_{i=1}^{|\mathcal{D}_\mathcal{T}|} \log \frac{1}{|\mathcal{D}_\mathcal{T}|} \sum_{j=1}^{|\mathcal{D}_\mathcal{T}|} \exp(\boldsymbol{z}_i^\top \boldsymbol{c}_j) + \log |\mathcal{D}_\mathcal{T}|$$

$$\ge -\frac{1}{|\mathcal{D}_\mathcal{T}|} \sum_{i=1}^{|\mathcal{D}_\mathcal{T}|} \boldsymbol{z}_i^\top \boldsymbol{c}_i + \frac{1}{|\mathcal{D}_\mathcal{T}|^2} \sum_{i=1}^{|\mathcal{D}_\mathcal{T}|} \sum_{j=1}^{|\mathcal{D}_\mathcal{T}|} \boldsymbol{z}_i^\top \boldsymbol{c}_j + \log |\mathcal{D}_\mathcal{T}|.$$

Since each prototype is expanded by $YY^\top$, there are $|\mathcal{C}_k|$ same vectors for each class $k$, where $\mathcal{C}_k$ denotes the set of samples of class $k$. Thus, we further have:

$$-\frac{1}{|\mathcal{D}_\mathcal{T}|}\sum_{i=1}^{|\mathcal{D}_\mathcal{T}|}\log\frac{\exp(\boldsymbol{z}_i^\top\boldsymbol{c}_i)}{\sum_{j=1}^{|\mathcal{D}_\mathcal{T}|}\exp(\boldsymbol{z}_i^\top\boldsymbol{c}_j)}\geq-\frac{1}{|\mathcal{D}_\mathcal{T}|}\sum_{i=1}^{|\mathcal{D}_\mathcal{T}|}\boldsymbol{z}_i^\top\boldsymbol{c}_i+\frac{1}{|\mathcal{D}_\mathcal{T}|}\sum_{i=1}^{|\mathcal{D}_\mathcal{T}|}\sum_{k=1}^{N_C}\frac{|\mathcal{C}_k|}{|\mathcal{D}_\mathcal{T}|}\boldsymbol{z}_i^\top\boldsymbol{c}_k$$

Similarly, for the second term, we can obtain the same results. In detail,

$$-\frac{1}{|\mathcal{D}_\mathcal{T}|}\sum_{i=1}^{|\mathcal{D}_\mathcal{T}|}\log\frac{\exp(\boldsymbol{c}_i^\top\boldsymbol{z}_i)}{\sum_{j=1}^{|\mathcal{D}_\mathcal{T}|}\exp(\boldsymbol{c}_i^\top\boldsymbol{z}_j)}=-\frac{1}{|\mathcal{D}_\mathcal{T}|}\sum_{i=1}^{|\mathcal{D}_\mathcal{T}|}\boldsymbol{c}_i^\top\boldsymbol{z}_i+\frac{1}{|\mathcal{D}_\mathcal{T}|}\sum_{i=1}^{|\mathcal{D}_\mathcal{T}|}\log\sum_{j=1}^{|\mathcal{D}_\mathcal{T}|}\exp(\boldsymbol{c}_i^\top\boldsymbol{z}_j)$$

$$=-\frac{1}{|\mathcal{D}_\mathcal{T}|}\sum_{i=1}^{|\mathcal{D}_\mathcal{T}|}\boldsymbol{c}_i^\top\boldsymbol{z}_i+\frac{1}{|\mathcal{D}_\mathcal{T}|}\sum_{i=1}^{|\mathcal{D}_\mathcal{T}|}\log\frac{1}{|\mathcal{D}_\mathcal{T}|}\sum_{j=1}^{|\mathcal{D}_\mathcal{T}|}\exp(\boldsymbol{c}_i^\top\boldsymbol{z}_j)+\log|\mathcal{D}_\mathcal{T}|$$

$$\geq-\frac{1}{|\mathcal{D}_\mathcal{T}|}\sum_{i=1}^{|\mathcal{D}_\mathcal{T}|}\boldsymbol{c}_i^\top\boldsymbol{z}_i+\frac{1}{|\mathcal{D}_\mathcal{T}|^2}\sum_{i=1}^{|\mathcal{D}_\mathcal{T}|}\sum_{j=1}^{|\mathcal{D}_\mathcal{T}|}\boldsymbol{c}_i^\top\boldsymbol{z}_j+\log|\mathcal{D}_\mathcal{T}|$$

$$=-\frac{1}{|\mathcal{D}_\mathcal{T}|}\sum_{i=1}^{|\mathcal{D}_\mathcal{T}|}\boldsymbol{c}_i^\top\boldsymbol{z}_i+\frac{1}{|\mathcal{D}_\mathcal{T}|}\sum_{k=1}^{N_C}\sum_{j=1}^{|\mathcal{D}_\mathcal{T}|}\frac{|\mathcal{C}_k|}{|\mathcal{D}_\mathcal{T}|}\boldsymbol{c}_k^\top\boldsymbol{z}_j.$$

Combining the two results, since the support data size is $n$, we then have:

$$\mathcal{L}_{\text{SCE}}\geq-\frac{2}{n}\sum_{i=1}^{n}\boldsymbol{z}_i^\top\boldsymbol{c}_i+\frac{2}{n}\sum_{i=1}^{n}\sum_{k=1}^{N_C}\frac{|\mathcal{C}_k|}{n}\boldsymbol{z}_i^\top\boldsymbol{c}_k.$$

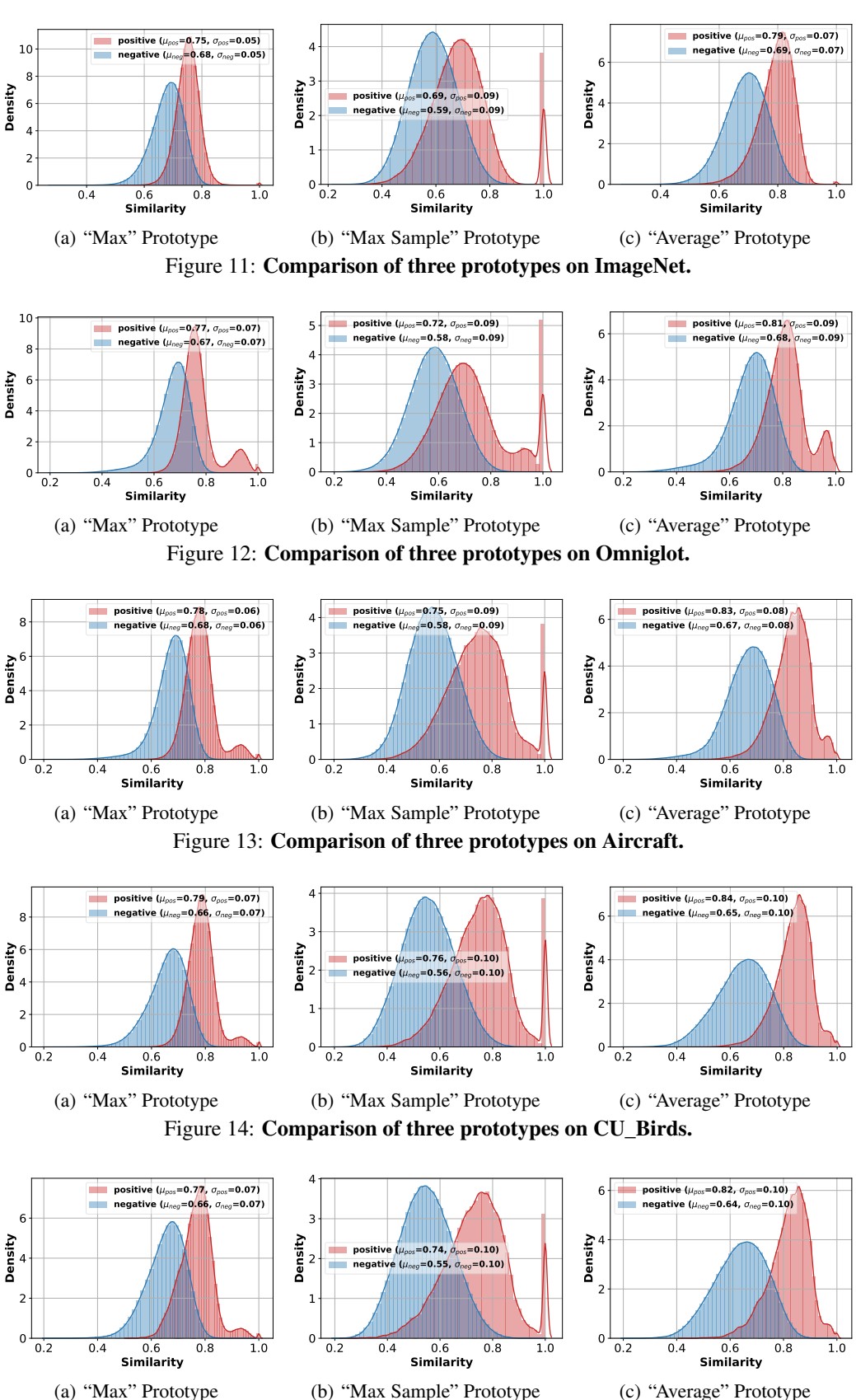

(a) "Max" Prototype      (b) "Max Sample" Prototype      (c) "Average" Prototype

Figure 11: **Comparison of three prototypes on ImageNet.**

(a) "Max" Prototype      (b) "Max Sample" Prototype      (c) "Average" Prototype

Figure 12: **Comparison of three prototypes on Omniglot.**

(a) "Max" Prototype      (b) "Max Sample" Prototype      (c) "Average" Prototype

Figure 13: **Comparison of three prototypes on Aircraft.**

(a) "Max" Prototype      (b) "Max Sample" Prototype      (c) "Average" Prototype

Figure 14: **Comparison of three prototypes on CU_Birds.**

(a) "Max" Prototype      (b) "Max Sample" Prototype      (c) "Average" Prototype

Figure 15: **Comparison of three prototypes on DTD.**

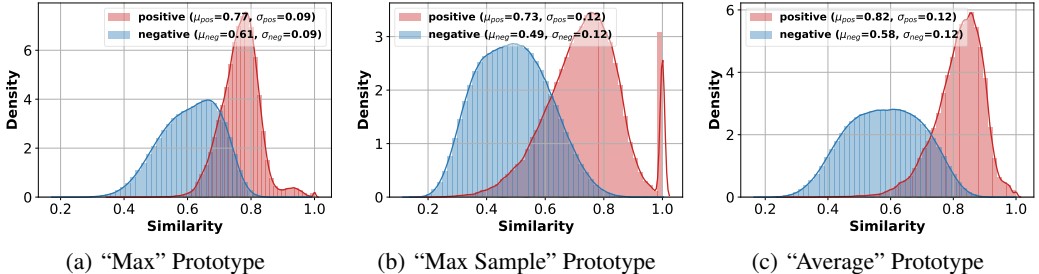

(a) "Max" Prototype      (b) "Max Sample" Prototype      (c) "Average" Prototype

Figure 16: **Comparison of three prototypes on Quick Draw.**

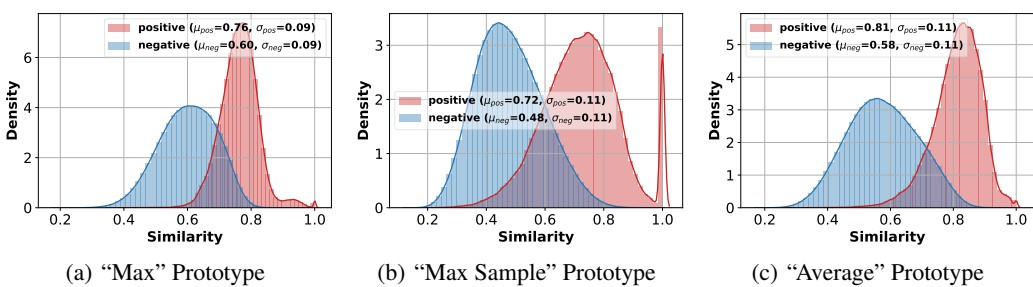

(a) "Max" Prototype      (b) "Max Sample" Prototype      (c) "Average" Prototype

Figure 17: **Comparison of three prototypes on Fungi.**

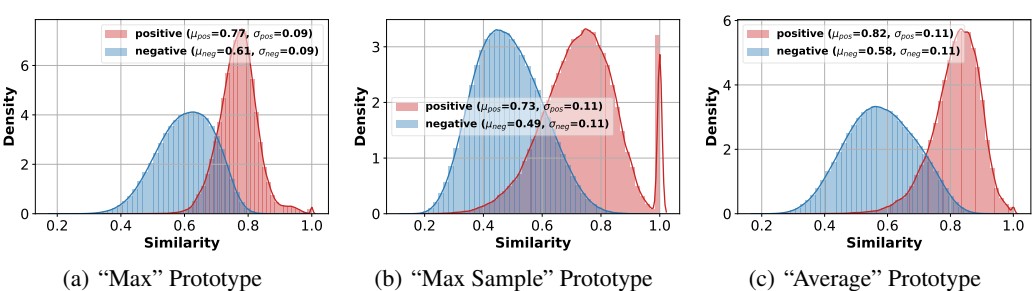

(a) "Max" Prototype      (b) "Max Sample" Prototype      (c) "Average" Prototype

Figure 18: **Comparison of three prototypes on VGG Flower.**

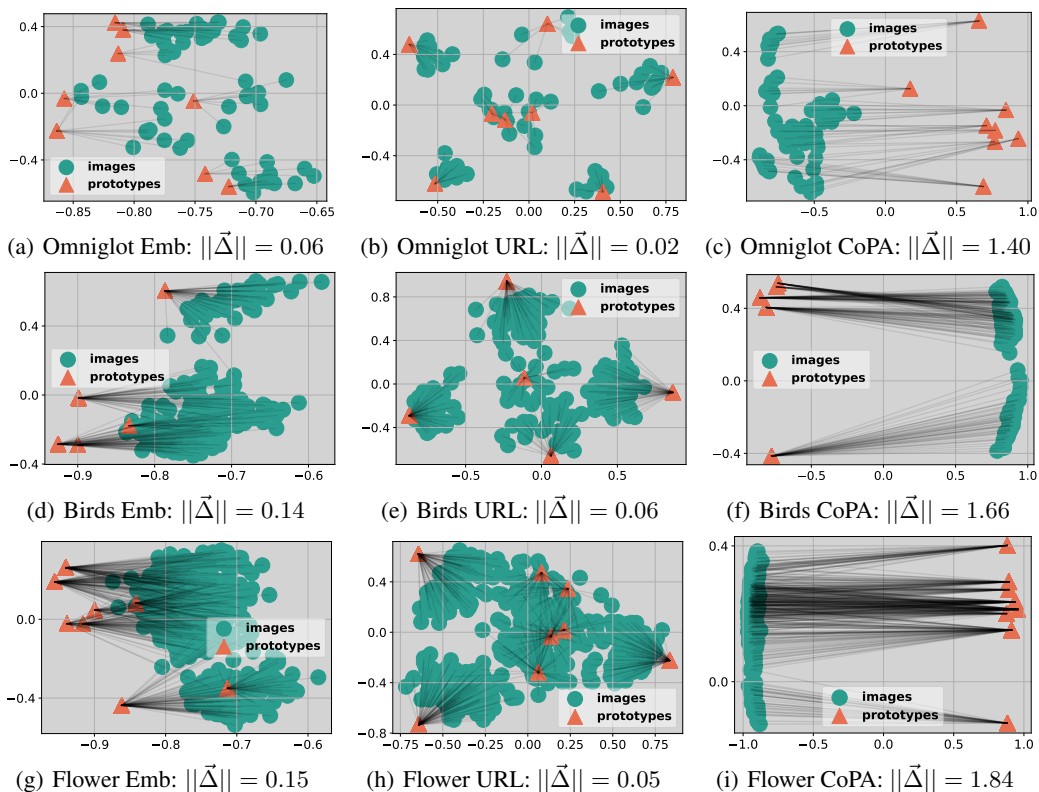

(a) Omniglot Emb: $||\vec{\Delta}|| = 0.06$   (b) Omniglot URL: $||\vec{\Delta}|| = 0.02$   (c) Omniglot CoPA: $||\vec{\Delta}|| = 1.40$

(d) Birds Emb: $||\vec{\Delta}|| = 0.14$   (e) Birds URL: $||\vec{\Delta}|| = 0.06$   (f) Birds CoPA: $||\vec{\Delta}|| = 1.66$

(g) Flower Emb: $||\vec{\Delta}|| = 0.15$   (h) Flower URL: $||\vec{\Delta}|| = 0.05$   (i) Flower CoPA: $||\vec{\Delta}|| = 1.84$

Figure 19: **"Modality" gaps between prototypes and images on some other datasets.**

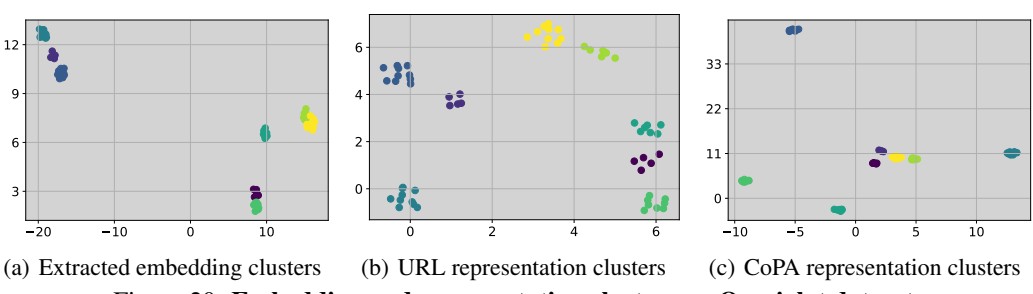

(a) Extracted embedding clusters   (b) URL representation clusters   (c) CoPA representation clusters

Figure 20: **Embedding and representation clusters on Omniglot dataset.**

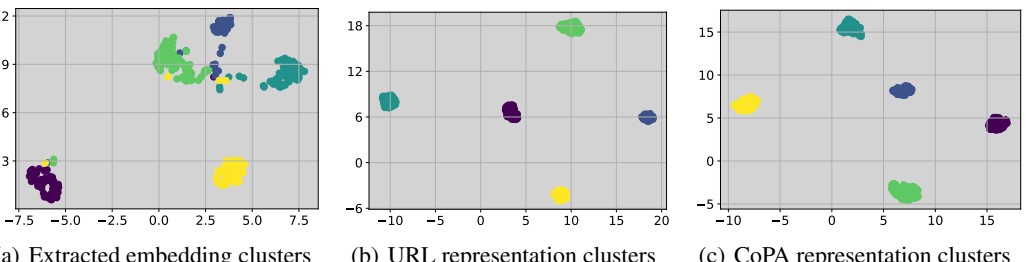

(a) Extracted embedding clusters   (b) URL representation clusters   (c) CoPA representation clusters

Figure 21: **Embedding and representation clusters on Aircraft dataset.**

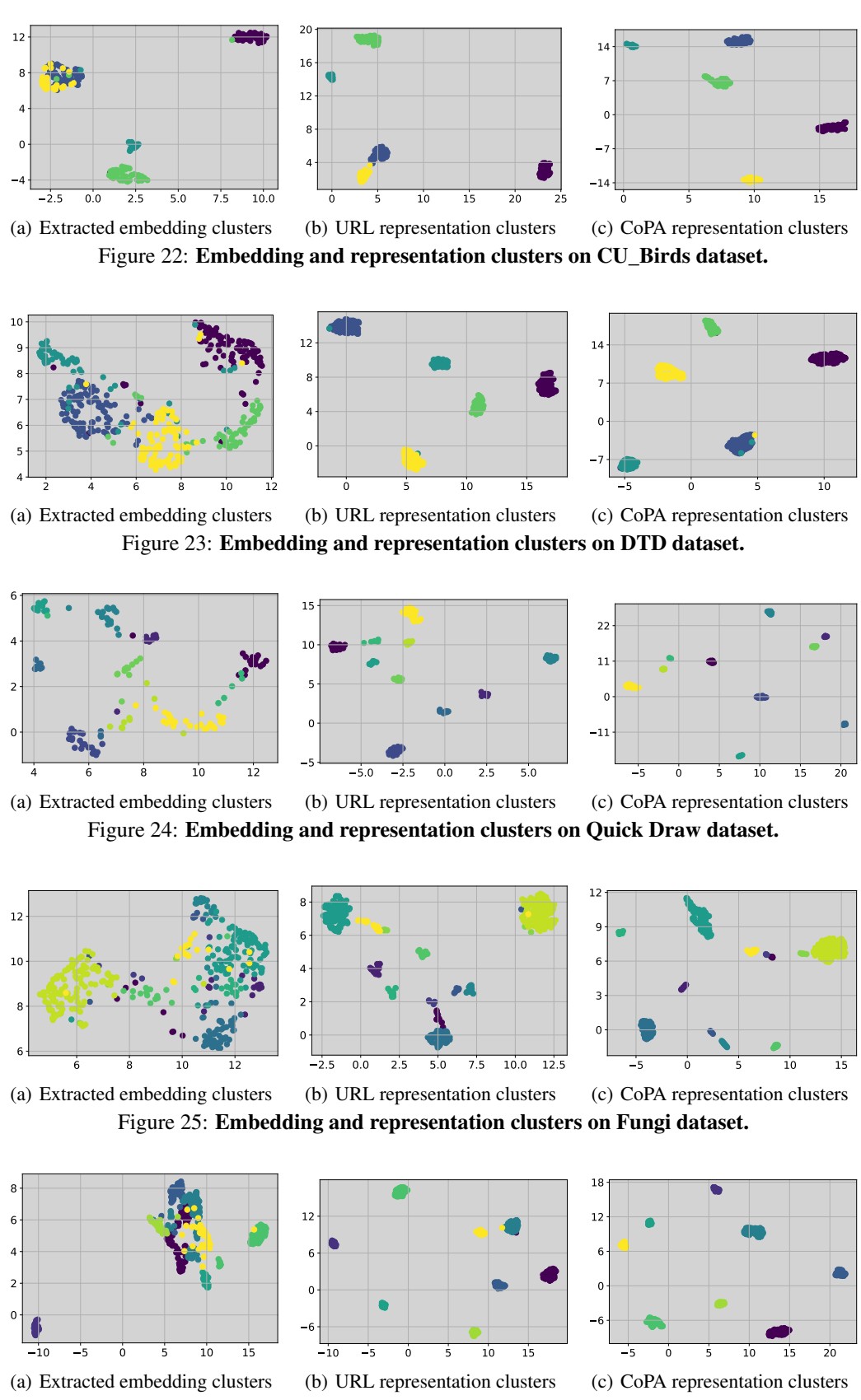

(a) Extracted embedding clusters     (b) URL representation clusters     (c) CoPA representation clusters

Figure 22: **Embedding and representation clusters on CU_Birds dataset.**

(a) Extracted embedding clusters     (b) URL representation clusters     (c) CoPA representation clusters

Figure 23: **Embedding and representation clusters on DTD dataset.**

(a) Extracted embedding clusters     (b) URL representation clusters     (c) CoPA representation clusters

Figure 24: **Embedding and representation clusters on Quick Draw dataset.**

(a) Extracted embedding clusters     (b) URL representation clusters     (c) CoPA representation clusters

Figure 25: **Embedding and representation clusters on Fungi dataset.**

(a) Extracted embedding clusters     (b) URL representation clusters     (c) CoPA representation clusters

Figure 26: **Embedding and representation clusters on VGG Flower dataset.**

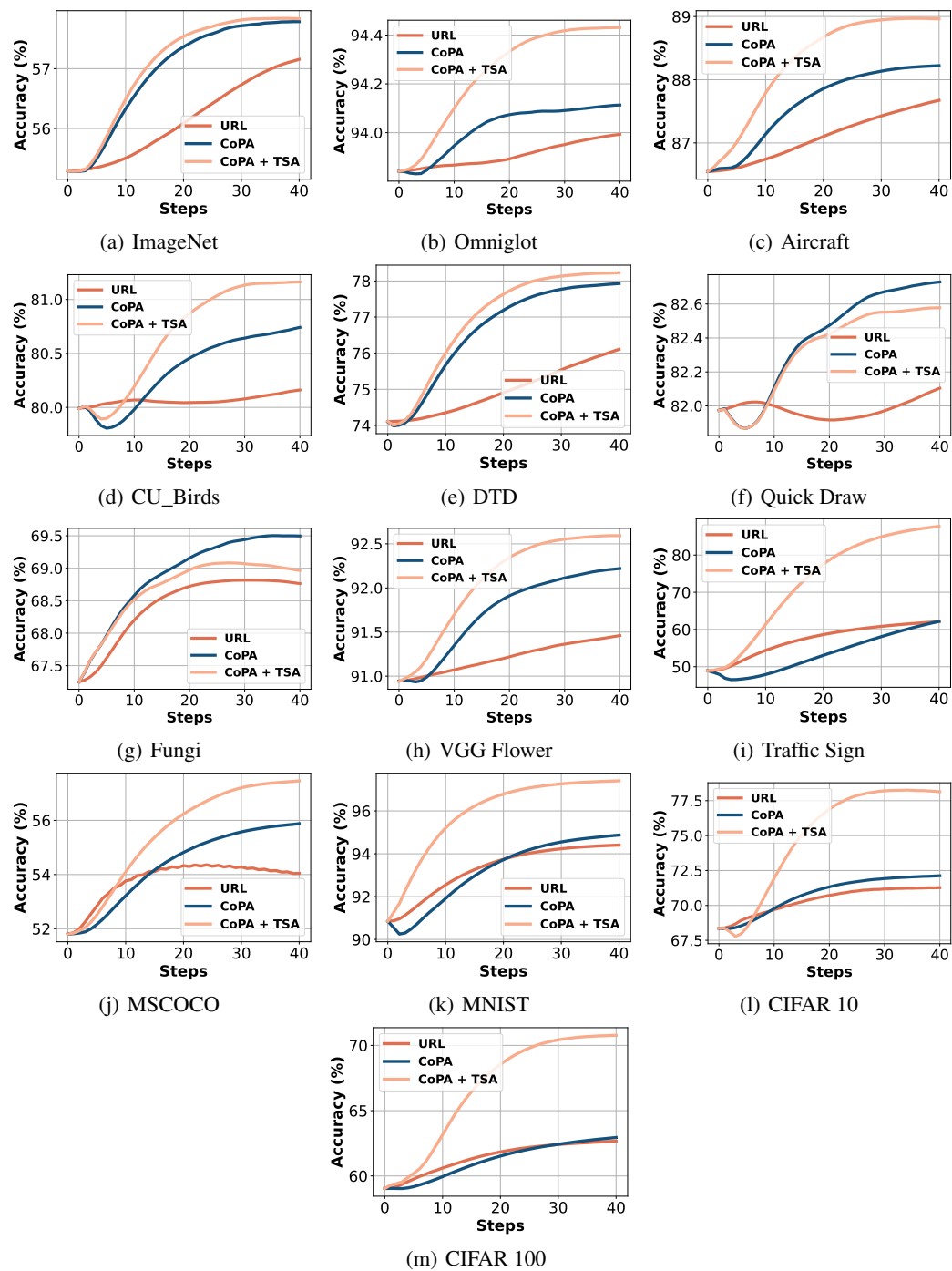

Figure 27: Generalization accuracy curves of Meta-Dataset with respect to the steps under 'Train on All Datasets' setting. As shown in figures, CoPA and CoPA + TSA evidently achieve better learning process and convergence performance compared with URL baseline.

