# OpenReview forum: "Mind the Gap Between Prototypes and Images in Cross-domain Finetuning"
_NeurIPS.cc/2024/Conference — NeurIPS 2024 poster_

### Official Review · Reviewer_PX84 · 2024-07-09

**Soundness:** 3
**Presentation:** 4
**Contribution:** 3
**Rating:** 7
**Confidence:** 4

**Summary:**

The authors of this paper found that in cross-domain few-shot classification, there naturally exists a gap, which resembles the modality gap, between the prototype and image instance embeddings. Thus, this paper proposes a simple yet effective method, contrastive prototype-image adaptation (CoPA), to adapt different transformations for prototypes and images similarly to CLIP by treating prototypes as text prompts. Extensive experiments on Meta-Dataset demonstrate that CoPA achieves state-of-the-art performance more efficiently. Meanwhile, further analyses also indicate that CoPA can learn better representation clusters, enlarge the gap, and achieve the minimum validation loss at the enlarged gap.

**Strengths:**

1. The paper is well written and visualized which is easy to read and understand.
2. The paper systematically analyzes and validates the modality gap between prototype and image instance embeddings in previous methods and proposes a simple yet effective method, contrastive prototype-image adaptation (CoPA), to adapt two different transformations for prototypes and image instances as done in CLIP via substituting text prompts with prototypes.
3. The experiments are sufficient and the results validate the theory in the previous section.
4. The supplementary material provides complete proof and results which are good.

**Weaknesses:**

One concern might be the novelty of this paper: the proposed CoPA is a classical method that has been proven to be effective in many previous works. However, I think the authors did a good job using old algorithms to solve new questions. Overall, I think this is a good paper that should be accepted.

**Questions:**

See weaknesses.

**Limitations:**

One potential limitation of our proposed CoPA method is the symmetric cross-entropy. Symmetric cross-entropy loss is commonly applied in contrastive learning. In contrastive learning, the batch size of the data samples is an important issue for better downstream tasks. Larger batch size usually means better downstream performance. We also notice this phenomenon in this paper. Although CoPA can still achieve relatively better performance, the performance tends to degrade when the size of the data batch becomes small. Thus, this constrains the application of CoPA to some extent.

---

> ### Author Rebuttal · Authors · 2024-08-05
>
> Thanks for your effort and time in reviewing our work. We appreciate a lot for your positive evaluation towards our work!
>
> For your concern about the novelty of our paper, we think that our contributions can be summarized as following:
>
> - We first noticed that prototypes play a similar role to the text prompts adopted in the multi-modal domain, and conjectured that the phenomenon that there is a gap between the prototype and image instance embeddings/representations may exist.
>
> - Then, we __empirically validated the gap between prototypes and image instances__. According to our empirical results, we found that
>     - applying a single transformation head as done in previous work may narrow the gap,
>     - the generalization performance can be improved by slightly enlarging the gap between the prototype and image instance embeddings.
>
> - To further explore the gap, we __conducted a series of theoretical analyses__. The results reveal that
>     - applying the same transformation potentially removes the discriminative information in gradients during the adaptation phase;
>     - the upper bound of the representation gap limits the change of the gap when applying a single transformation head.
>
> - Based on the observations and inspired by the previous work, we followed CLIP and proposed to use two decoupled transformations and substitute the text prompts in the SCE loss with prototypes to preserve the gap between the prototypes and image instances. Thus, we proposed an efficient and effective CFC adaptation method, CoPA.
>
> - According to our experimental results, CoPA is able to
>     - improve the generalization performance without hurting the efficiency，
>     - preserve the gap between prototype and image instance representations,
>     - explore the optimal distributions (representation space) to align the representations for better performance.
>
> A potential novelty of our proposed CoPA method is that __CoPA provides a new perspective of representation learning__. Specifically, we can consider manually designing some kinds of data that contain different levels of information from the original data. Then, by performing representation alignment based on the flexible hypothesis space, we can learn better representations and achieve better performance without extra data. In this way, we can take advantage of the existing data more sufficiently and effectively.

---

> ### Author Response · Authors · 2024-08-09
> **Do you have any further concerns or questions?**
>
> Dear Reviewer PX84,
>
> Thanks again for your time and efforts in reviewing our work. We appreciate your valuable and insightful comments on our work. We believe your expertise will help improve the quality of our work.
>
> This is a gentle reminder that the Discussion period will end on 13 Aug 11:59 pm AoE. Would you mind checking our responses to your concerns and questions and confirming whether you have any other questions? We are glad to answer any further questions you may have before the deadline.
>
> Best regards,
>
> Authors

---

> > ### Comment · Reviewer_PX84 · 2024-08-12
> >
> > Thanks for your response,
> >
> > I have no more questions, and will not reduce the score. Good luck!

---

> > > ### Author Response · Authors · 2024-08-12
> > > **Thanks a lot for your support!**
> > >
> > > Dear Reviewer PX84,
> > >
> > > Thanks for maintaining 7! We appreciate a lot for your positive comments and support to our work!
> > >
> > > Best wishes,
> > >
> > > Authors

---

### Official Review · Reviewer_eGag · 2024-07-10

**Soundness:** 2
**Presentation:** 2
**Contribution:** 2
**Rating:** 4
**Confidence:** 3

**Summary:**

This paper claims that there exists gap between prototypes and image instance embeddings in cross-domain few-shot classification models.
URL[29], a representative work of cross-domain few-shot classification, proposes to fast fine-tune linear classifier on top of a frozen back bone with the nearest centroid classifier loss.
In URL, the embeddings of an image instance encode the instance-level information from the given image, while the embedding of a prototype contain some abstract and high-level information among image instance.
This paper proposes Contrastive Prototype-image Adaptation (CoPA), to adapt different transformations for prototypes and image embeddings to adjust the modality gap.

**Strengths:**

-  This paper show the existence of gaps between prototype and image embedding.
- CoPA achieves the state-of-the-art performance on Meta-dataset.
- CoPA can enlarge the gap to achieve the global minimum validation loss and learn more compact representation clusters.

**Weaknesses:**

The CoPA looks only improving the performance by adding complexity of models and losses.
- CoPA have two linear heads that is more complicated than URL. This paper argues that discriminative information of gradients is removed when applying the same transformation. This claim seems to be about that when one common classifier is used, gradient does not consider the model parameters of two linear classifier (Discission of Theorem 3.1) However, it is obvious that the model that uses one linear classifier does not consider difference of two classifiers in gradients.
- There is no clear reason that enlarging the gap between the prototypes and instance embeddings improve performance (though is empirically reducing the validation loss.)
CoPA is based on such a uncertain phenomenon.
- Also CoPA have a temperature parameter $\gamma$ in the symmetric cross entropy, meanwhile the NCC objective does not. It seems to be unfair, as the temperature influence the gap [32].

**Questions:**

This paper discuss the gap between prototype of class c, and image instances. However, I could not understand why such large gap exist. Since the mean vector is the average of image instances, the prototypes should be within the convex full of image instances. Thus, I feel strange that instances and prototypes are in the separate regions as shown in Fig.1 .

The number of samples per classes will influence the gap $\Delta$. Could you justify this issue?

---

> ### Author Rebuttal · Authors · 2024-08-06
>
> We post our responses regarding your concerns in the following:
> >__Weakness 1__: CoPA has two ... one linear classifier does not consider difference of two classifiers in gradients.
>
> __Answer__: According to your concern, you also believe that a single transformation cannot consider the difference of gradients. That is what we wanted to highlight. The shared transformation is incompetent in CFC adaptation tasks.
>
> In fact, __the improvements of CoPA cannot be simply attributed to the complexity of models and loss__.
>
> In this paper,
> - we first noticed that prototypes play a similar role to the text prompts in CLIP.
> - Then, we empirically demonstrated that there exists a gap between the image and prototype embeddings, which is similar to the gap found in CLIP, and slightly enlarging the gap helps improve the generalization performance.
> - According to further analyses, we found that the gap may be narrowed since applying a single transformation may damage the discriminative information in gradients.
>
> This naturally motivated us to consider using two different transformations and substituting the text prompts with prototypes, as done in CLIP, to treat prototypes and images independently and perform representation alignment for better generalization performance. These points are highly appreciated by other reviewers.
>
> Per your concern about complexity, as we have discussed in Section 5.2 (Line 330-338), __applying decoupled transformations does increase the complexity of the hypothesis space, but it reduces the approximation error and is more flexible in representation learning (preserving the discriminative information)__. Meanwhile, we conducted several ablation studies in Appendix F3 & F5. As shown in Table 9, we perform URL with an MLP head (two 512$\times$512 layers). However, the performance drops evidently. This indicates that __it is the flexible hypothesis space, instead of the model capacity, that facilitates the generalization performance__. Besides, as shown in Table 10, although equipped with SCE loss (same as CoPA), URL fails to achieve similar performance to CoPA. This reveals that __merely using SCE is insufficient for better generalization performance__. Thus, the success of CoPA cannot be simply attributed to the complexity of the model and loss.
>
> >__Weakness 2__: There is no clear reason that ... such a uncertain phenomenon.
>
> __Answer__: Per your concern, according to the paper, we noticed that prototypes play a similar role to the text prompts. Specifically, a prototype is the "common features" of a class of images, and a text prompt summarizes the "common sense" of a class of images. Thus, some semantic differences should exist between images and prototypes. According to empirical results, a gap does exists between prototypes and images. Such a gap describes the differences between prototypes and instances.
>
> Meanwhile, as demonstrated in CLIP, for good generalization performance, representations of images and texts should be aligned. Thus, in Section 5.2, we argued that the essence of the gap enlargement is a kind of alignment between prototype and image instance representations, in which the optimal distributions of prototype and instance representations are explored. For a pre-trained model, where representations are well aligned, better generalization performance can be obtained.
>
> >__Weakness 3__: Also CoPA have a temperature parameter ... seems to be unfair...
>
> __Answer__: Per your concern, the role of the temperature coefficient is tuning how concentrated the features are in the representation space. __In fact, in practice, the NCC objective also adopts temperature coefficient, though it is not mentioned in the paper__.
>
> In the source code of URL (see Line 149 in `./models/losses.py` file of our submitted supplementary file), a temperature coefficient 10 is multiplied to the logits. When reproducing URL, we __adopted all original settings for fairness__.
>
> >__Question 1__: This paper discuss the gap ... Thus, I feel strange that instances and prototypes are in the separate regions as shown in Fig.1. The number of samples per classes will influence the gap $\Delta$. Could you justify this issue?
>
> __Answer__: Per your concern about the gap, we discuss it from two different angles. Firstly, if __representations are not normalized__, the prototype is within the convex of instances and influenced by the number of samples. Specifically, when the number of samples in each class is equal, the gap is 0.
>
> However, according to [32], the calculation of the gap is based on __normalized embeddings/representations__ (we have highlighted this in line 174-175). Here, we take the embedding gap as an example.
>
> Consider a set of __unnormalized enbeddings__ $\{f\_{\phi^*}(\boldsymbol{x}\_i)\}\_{i=1}^{|\mathcal{D}\_{\mathcal{T}}|}$, the prototype for class $c$ is $\boldsymbol{c}\_{c}=\frac{1}{|\mathcal{C}\_c|}\sum\_{f\_{\phi^*}(\boldsymbol{x})\in\mathcal{C}\_c}f\_{\phi^*}(\boldsymbol{x})$, where $\mathcal{C}\_c=\\{f\_{\phi^*}(\boldsymbol{x}\_i)|y\_i=c\\}$. Then, the gap vector is calculated as $\vec{\Delta}=\frac{1}{|\mathcal{D}\_{\mathcal{T}}|}\sum\_{i=1}^{\mathcal{D}\_{\mathcal{T}}}\frac{f\_{\phi^*}(\boldsymbol{x}\_i)}{||f\_{\phi^*}(\boldsymbol{x}\_i)||\_2}-\frac{1}{N\_C}\sum\_{j=1}^{N\_C}\frac{\boldsymbol{c}\_j}{||\boldsymbol{c}\_j||\_2}$ (see P7 of [32]). It is easy to see that the embeddings will be normalized when calculating the gap. In this case, __the prototype may not be in the convex of images, and the relationship between the number of samples of each class and the gap is undetermined__.
>
> To further solve your concern, we checked the gap between prototype and instance embeddings under the 'vary-way 5-shot' setting, where the number of samples of each class is equal. It is easy to see that __the gap still exists__.
> |Methods|ImageNet|Omniglot|Aircraft|Birds|DTD|QuickDraw|Fungi|Flower|
> |---|---|---|---|---|---|---|---|---|
> |__Gap__|0.15|0.10|0.10|0.10|0.11|0.12|0.12|0.12|

---

> ### Author Response · Authors · 2024-08-09
> **Do you have any further concerns or questions?**
>
> Dear Reviewer eGag,
>
> Thanks again for your time and efforts in reviewing our work.
>
> This is a gentle reminder that __the Discussion period will end on 13 Aug 11:59 pm AoE__. Would you mind checking our responses to your concerns and questions and confirming whether you have any other questions? We are glad to answer any further questions you may have before the deadline.
>
> Best regards,
>
> Authors

---

> ### Author Response · Authors · 2024-08-12
> **Do our responses solve your concerns?**
>
> Dear Reviewer eGag,
>
> Thanks again for your time and efforts in reviewing our work. Do you still have any other concerns or questions? We are glad to answer any further questions you may have before the deadline.
>
> Best regards,
>
> Authors

---

> ### Author Response · Authors · 2024-08-13
> **One day remaining for the discussion period**
>
> Dear Reviewer eGag,
>
> Thanks again for your time in reviewing our work.
>
> This is a gentle reminder that the Discussion period will end on __13 Aug 11:59 pm AoE__. There is __only one day remaining__. Would you mind checking our responses to your concerns and questions and confirming whether you have any other questions? We are glad to answer any further questions you may have before the deadline.
>
> Best regards,
>
> Authors

---

> > ### Comment · Reviewer_eGag · 2024-08-13
> >
> > Thank you for the response.
> >
> > Some of my concerns have been solved. For examples, it is not strange that gap exist if the instances and centroids are normalized.
> >
> > I understood that authors followed the observation of [32], but why enlarging the gap between centroid and instances leads to improving the performance is not clear for me. Also, why different model parameters lead to enlarge the gap is not clear. I understand the discriminative information means the difference of gradients of model parameters for prototypes and instance embeddings. However, flexible model can both possible to enlarge or reduce the gap.
> >
> > I understand that this work is inspired form the modality gap [32] of CLIP, and thus using the symmetric cross-entropy of CLIP. I have noticed that Table 10 of Appendix compare loss functions. However, this paper does not explain why using only two linear projections on NCC loss does not work well and how the symmetric cross entropy of CLIP solves this problem.

---

> > > ### Author Response · Authors · 2024-08-13
> > > **Further responses to your new concerns**
> > >
> > > Thanks for your reply. We are glad that our responses partially solved your concerns.
> > >
> > > For your concern, the enlargment of the gap and the generalization performance are still open problems in the literatures. Currently, reseachers, including us, only empirically observe such phenomena, and find that generalization performance (zero-shot, few-shot etc.) is associated with the gap. All of us try to figure out and explain these problems from different perspectives.
> > >
> > > In the following, we will try to address your new concerns, based on the insightful discussions with Reviewer Ms7f.
> > >
> > > > Gap enlargement & generalization performance
> > >
> > > For your concern about why enlarging the gap improves the generalization performance, __inspired by the [comments from Reviewer Ms7f](https://openreview.net/forum?id=JWLiK3kKWQ&noteId=HRvuDHQQfk#:~:text=Based%20on%20the,different%20transformation%20heads)__, we find that overfitting and regularization may be a good perspective to explain. Specifically, __enlarging the gap reaches a balance between learning discriminative representations and achieveing better generalization performance__.
> > >
> > > According to Fig. 3(a), it is easy to observe that the enlargement of the gap is constrained in a small range. Two cases are worth discussion here:
> > > - If the gap is narrowed, the similarity between prototypes and instances increases. However, the validation loss also increases. __Such a phenomenon indicates that narrowing the gap may result in overfitting__. As a demonstration, we can also observe the overfitting phenomenon happens in some cases from Fig. 27 (f), (g), and (j) (__the orange line__).
> > > - If the gap is infinitely enlarged, the validation loss will also increase since the similarity between the prototypes and instances decreases.
> > >
> > > Thus, from this perspective, __the enlargement of the gap can be seen as a regularization that drives the model to explore a balance between learning better representations and achieving better generalization performance__.
> > >
> > > > Why NCC + 2 heads does not work well and how symmetric cross-entropy loss address this issue.
> > >
> > > Based on the above findings, we find that we can give a more reasonable explaination to this issue as well. For your concern about why NCC loss along with 2 different transformation heads does not work well, we further conduct an experiment here to see if 2Heads+NCC has an overfitting issue. For simplicity, we only applied two different transformation heads on URL method with the random seed 42 and ImageNet dataset (due to the time constraint, but we will include more experiments for this in the updated version of our paper).
> > >
> > > |Methods|Train Acc|Test Acc|
> > > |---|---|---|
> > > |__URL+2heads+NCC__|94.93|56.67|
> > > |__CoPA__|90.87|57.92|
> > >
> > > According to the results, in the "URL+2heads+NCC" case, we observe __higher training accuracy and lower test accuracy__ compared to CoPA. __This is a phenomenon of overfitting__ (compared to CoPA). However, applying the symmetric cross-entropy loss alleviates such a phenomenon to some extent (see the training and test accuracy of CoPA).
> > >
> > > Thus, based on the explanation above (thank Reviewer Ms7f for this valuable insight!), __the symmetric cross-entropy loss itself might implicitly regularize the model during the learning process and achieve a balance between the discriminability and the transferability of the learned representations__.
> > >
> > > In summary, we think that these explanations are interesting, and we will merge these discussions in our updated version.

---

> ### Author Response · Authors · 2024-08-14
> **Look forward to your feedback**
>
> Dear Reviewer eGag,
>
> Thanks for your time and efforts in reviewing our work.
>
> We have posted responses to your new concerns. Would you mind checking the responses and confirming whether the updated responses solve your concerns? We are glad to answer any concerns and questions you may have before the deadline.
>
> __As a gentle reminder, the discussion period will end in about 12 hours__. We hope to resolve your concerns about our work.
>
> Moreover, if your concerns have been resolved, would you mind reconsidering your rating of our work based on the updated understanding?
>
> Best wishes,
>
> Authors

---

### Official Review · Reviewer_Ms7f · 2024-07-13

**Soundness:** 3
**Presentation:** 3
**Contribution:** 3
**Rating:** 6
**Confidence:** 3

**Summary:**

This paper investigates the gap between the prototype and image instance embeddings under the setting of cross-domain few-shot classification. It shows that applying the same transformation to these embeddings will shrink their gap and constrain the exploration of optimal representation distributions. Based on a series of empirical and theoretical analyses, this paper proposes a prototype-image contrastive learning adaptation method, namely CoPA. The experimental results on Meta-Dataset indicates the effectiveness of the proposed method.

**Strengths:**

1.	This paper is well organized, easy to follow and free of typos.
2.	Extensive empirical and theoretical analyses are provided, so the proposed contrastive learning method seems to be technically sound.
3.	The implementation details are clearly stated, the algorithm and source code are provided, ensuring the reproducibility of the method.
4.	Experiments and ablation studies are adequate, the results under different experimental settings are convincing and promising.

**Weaknesses:**

1.	For the empirical analysis in Section 3.2, there is an interesting phenomenon that “appropriately enlarging the gap between the prototypes and image instances contributes to achieving better generalization performance”. It may be better to provide some discussions of the reasons for these these observed results (Figure 3(a)) to explain the influence of “enlarging the gap” on the “generalization ability”.
2.	The competing methods used in experiments appear to be somewhat outdated. The main baseline URL is published in 2021. It may be better to compare with some recently proposed adaptation-based methods.
3.	Some equation numbers are missing. Some references are not correctly formatted, the conference or journal names as well as page numbers are missing.

**Questions:**

It is a good paper, I personally like this paper and just have two small question:

1.	In Theorem 3.1, optimizing the first term in the equation between the 206th and 207th rows requires to maximize the similarities between instances and their corresponding protypes. So, would enlarging the protype-image gap negatively affect this optimization process?

2.	According to the equations between the 218th and 219th rows, I can understand the differences of the first term in these two equations are removed, when the shared transformation is adopted. But why this may lead to the drop of “the discriminative information in gradients”? Could the authors explain this more clearly?

**Limitations:**

The authors have discussed the limitations.

---

> ### Author Rebuttal · Authors · 2024-08-05
>
> Thanks for your effort and time in reviewing our work. We appreciate a lot for your insightful and valuable comments towards our work. Now, we post our responses regarding your concerns in the following:
> > __Weakness 1__: For the empirical analysis in Section 3.2, there is an interesting phenomenon that “appropriately enlarging the gap between the prototypes and image ...”. It may be better to provide some discussions of the reasons for these these observed results (Figure 3(a)) to explain the influence of “enlarging the gap” on the “generalization ability”.
>
> __Answer__: Thanks for your interesting question. As we have mentioned in the paper, we found that the prototypes play the similar role to the text prompts used in multi-modal domains. __Specifically, a prototype is the "common features" shared within a class of images, and the text prompt summarizes the "common sense" of a class of images__. From this perspective, there should exist some kind of semantic differences between prototypes/text prompts and image instances. As shown in Fig. 1 and other visualization results in Appendix B, there do exist a gap between the prototype and image instance embeddings. Such a gap can be seen as the difference between two sets of data.
>
> According to previous works, such as CLIP, to achieve good performance, the representations of images and texts should be aligned. Thus, as we argued in Section 5.2, we think __the essence of the gap enlargement is a kind of alignment between prototype and image instance representations, in which the optimal distributions of prototype and instance representations are explored__. Thus, for a pretrained model, where representations are well aligned, better generalization performance can be obtained.
>
> >__Weakness 2__: The competing methods used in experiments appear to be somewhat outdated. The main baseline URL is published in 2021. It may be better to compare with some recently proposed adaptation-based methods.
>
> __Answer__: Thanks for your suggestion. In fact, we have compared CoPA with some recent representative adaptation-based baselines in our paper. Specifically, we considered TSA (CVPR 2022) and TA$^2$-Net (ICCV 2023), which achieved better empirical performance by leveraging extra adaptation learning modules, in Table 1 & 2. According to the results, CoPA outperforms these baselines evidently. We will continue to review other works and consider adding them as references in our updated version.
>
> >__Weakness 3__: Some equation numbers are missing. Some references are not correctly formatted, the conference or journal names as well as page numbers are missing.
>
> __Answer__: Thanks for your suggestions. We will add equation numbers to all equations and modify the reference format in our updated versions.
>
>
> >__Question 1__: In Theorem 3.1, optimizing the first term in the equation between the 206th and 207th rows requires to maximize the similarities between instances and their corresponding protypes. So, would enlarging the protype-image gap negatively affect this optimization process?
>
> __Answer__: Thanks for your insightful question. We would like to discuss this question via our existing empirical results. First of all, according to Fig. 3(a), it is easy to observe that __the gap cannot be enlarged infinitely__ and __the enlargement is limited in a small range__. The validation loss tends to increase when the gap is larger than a threshold. This is reasonable since significantly enlarging the gap, as you mentioned, reduces the similarities between instances and their prototypes. Meanwhile, narrowing the gap (increasing the similarity) also results in the increase of validation loss, as excessively maximizing the similarity may lead to generalization problems (e.g., overfitting).
>
> Moreover, according to Fig. 5(c), the minimal validation loss is achieved at the learned representation gap. In this case, both enlarging and narrowing the gap increase the validation loss. Since SCE can align different sets of representations and preserve the gap between two sets of data, we argued that __the essence of the enlargement of the gap between image and prototype representations is exploring the optimal function space for representation alignment__ (see Section 5.2). Thus, we think such a slight enlargement will not negatively affect the optimization process.
>
> This question is interesting. We will add this discussion in our updated version.
>
>
> >__Question 2__: According to the equations between the 218th and 219th rows, I can understand the differences of the first term in these two equations are removed, when the shared transformation is adopted. But why this may lead to the drop of “the discriminative information in gradients”? Could the authors explain this more clearly?
>
> __Answer__: Thanks for your question. When two transformations are applied, the gradients for image instance and prototype transformations are respectively calculated with the equations mentioned in the paper. As you can see, the two gradients are different.
>
> However, when applying a __single__ transformation ($\Theta_{\rm I}=\Theta_{\rm P}=\Theta$), the gradient of $\Theta$ is $\nabla\_{\Theta}\mathcal{L}(\Theta)=-\frac{2}{|\mathcal{D}\_{\mathcal{T}}|}\Theta^{\top}f\_{\phi^*}(\boldsymbol{X})^{\top}YY^{\top}f\_{\phi^*}(\boldsymbol{X}) + \frac{2\alpha}{|\mathcal{D}\_{\mathcal{T}}|}\Theta^{\top}f\_{\phi^*}(\boldsymbol{X})^{\top}f\_{\phi^*}(\boldsymbol{X})$, where
> - the first term is the combination of the first terms of $\nabla_{\Theta_{\rm P}}\mathcal{L}(\Theta_{\rm P}, \Theta_{\rm I})$ and $\nabla_{\Theta_{\rm I}}\mathcal{L}(\Theta_{\rm P}, \Theta_{\rm I})$ with $\Theta_{\rm P}=\Theta_{\rm I}=\Theta$,
> - the second term is the second term of $\nabla_{\Theta_{\rm I}}\mathcal{L}(\Theta_{\rm P}, \Theta_{\rm I})$ with $\Theta_{\rm I}=\Theta$.
>
> It is easy to find that the discriminative information of gradients is damaged since the gradient information regarding prototypes and image instances is mixed.

---

> > ### Comment · Reviewer_Ms7f · 2024-08-12
> >
> > Thank you for your response!
> >
> > Based on the authors' reply, I think that the "enlarging the gap between prototype and image instance representations" is more like a regularization that prevent the image representations from overfitting to the class semantic information rather than an alignment between text and image features as in CLIP. Specifically, in my own opinion, the enlargement operation in this paper is to balance the "discriminability" and the "transferability" of the learned representations by using two different transformation heads, while the "alignment" is to pull closer the features with the same semantics and push away those with different semantics. So it maybe better for the authors to provide more clear explanations about this. Nevertheless, I personally like this paper and will maintain the rating score of "weak accept".

---

> > > ### Author Response · Authors · 2024-08-12
> > > **Thanks for your support!**
> > >
> > > Dear Reviewer Ms7f,
> > >
> > > Thanks a lot for your appreciation and support to our work! The opinion you proposed is interesting and can be used to explain CoPA from another perspective.
> > >
> > > We agree with your opinion that there exists overfitting in previous works. Two aspects can demonstrate this.
> > > - On the one hand, as we have mentioned, closing the gap increases the similarity between the prototypes and image instances. However, according to Fig. 3(a), the validation loss also increases. This indicates the overfitting phenomenon to some extent.
> > > - On the other hand, we can also observe the overfitting phenomenon happens in some cases from Fig. 27 (f), (g), and (j) (__the orange line__).
> > >
> > > Both phenomena demonstrate the existence of the overfitting problem in previous work (i.e. URL). Thus, from this perspective, enlarging the gap can be viewed as __exploring a balance between learning better representations ("discriminability") and achieving better generalization performance ("transferability")__. According to the data clusters (Fig. 5(b)) and Fig. 27 (__the blue line__), our proposed CoPA does __achieve such a balance__.
> > >
> > > We will merge this discussion in our updated version.

---

> ### Author Response · Authors · 2024-08-09
> **Do you have any further concerns or questions?**
>
> Dear Reviewer Ms7f,
>
> Thanks again for your time and efforts in reviewing our work. We appreciate your valuable and insightful comments on our work. We believe your expertise will help improve the quality of our work.
>
> This is a gentle reminder that __the Discussion period will end on 13 Aug 11:59 pm AoE__. Would you mind checking our responses to your concerns and questions and confirming whether you have any other questions? We are glad to answer any further questions you may have before the deadline.
>
> Best regards,
>
> Authors

---

### Decision · Program_Chairs · 2024-09-25

**Decision:**

Accept (poster)

**Comment:**

The paper received 6/4/7 ratings. The majority of the concerns come from the validity of the major claim of the paper. The authors provided a rebuttal. R1 is convinced with the explanation, while R2 still feels it questionable. R2 also mentioned that he can go either way with the paper's decision. After reading the paper, reviews, and rebuttal, the AC recommends accepting the paper. The findings and solution will benefit the community.